# GOOD: Geometry-guided Out-of-Distribution Modeling for Open-set Test-time Adaptation in Point Cloud Semantic Segmentation

**Tianpei Zou**[1,2], **Guo Yu**[1,3], **Ya Wu**[4], **Fan Lu**[1], **Zhongcong Xu**[5],
**Zhang Bo**[3], **Ziqiao Wang**[1], **Sanqing Qu**[1†], **Guang Chen**[1,2,3†*]

[1] Tongji University  [2] Shanghai Innovation Institution  [3] Shanghai Westwell Technology Co., Ltd.
[4] CNNC Equipment Technology Research (Shanghai) Co., Ltd.  [5] ByteDance Seed
[†] Corresponding author

## Abstract

Open-set Test-time Adaptation (OSTTA) has been introduced to address the challenges of both online model optimization and open-set recognition. Despite the demonstrated success of OSTTA methodologies in 2D image recognition, their application to 3D point cloud semantic segmentation is still hindered by the complexities of point cloud data, particularly the imbalance between known (in-distribution, ID) and unknown (out-of-distribution, OOD) data, where known samples dominate and unknown instances are often sparse or even absent. In this paper, we propose a simple yet effective strategy, termed Geometry-guided Out-of-Distribution Modeling (GOOD), specifically designed to address OSTTA for 3D point cloud semantic segmentation. Technically, we first leverage geometric priors to cluster the point cloud into superpoints, thereby mitigating the numerical disparity between individual points and providing a more structured data representation. Then, we introduce a novel confidence metric to effectively distinguish between known and unknown superpoints. Additionally, prototype-based representations are integrated to enhance the discrimination between ID and OOD regions, facilitating robust segmentation. We validate the efficacy of GOOD across four benchmark datasets. Remarkably, on the Synth4D to SemanticKITTI task, GOOD outperforms HGL by 1.93%, 8.99%, and 7.91% in mIoU, AUROC, and FPR95, respectively.

## 1 Introduction

3D point cloud semantic segmentation is crucial for scene understanding in many applications like autonomous driving (Wu et al., 2018; Choy et al., 2019) and robotics (Balsiger et al., 2019; Liu et al., 2023). However, deep neural network (DNN)-based models often experience performance degradation when exposed to unseen or a different target distribution than the source training data (Li et al., 2023a; Duan et al., 2024). Test-time Adaptation (TTA) has emerged as a promising technique to mitigate these challenges by leveraging unlabeled data to update models dynamically for deployment scenarios. Yet, most TTA methods focus primarily on addressing covariate shifts (distributional differences in known classes), often neglecting semantic shift scenarios in which the target data includes novel categories absent from the training data. This oversight may pose significant risks in security-critical applications, as models may misinterpret or fail to recognize unknown objects, as illustrated in Fig. 1 a) and b).

Recent efforts in 2D image recognition (Li et al., 2023b; Lee et al., 2023; Gao et al., 2024; 2023; Yu et al., 2024; Schlachter & Yang, 2024) have introduced Open-set Test-time Adaptation (OSTTA) methods for online model optimization and open-set detection in generalized cases involving out-of-distribution (OOD) categories. However, directly adapting these OSTTA methods to 3D point cloud semantic segmentation is suboptimal and challenging. As shown in Fig.1 c) and g), combining TTA methods designed for 3D point cloud segmentation (TTA-3DSeg), such as GIPSO (Saltori et al.,

---

[*] Project leader, supervised the work and defined the conceptualization.

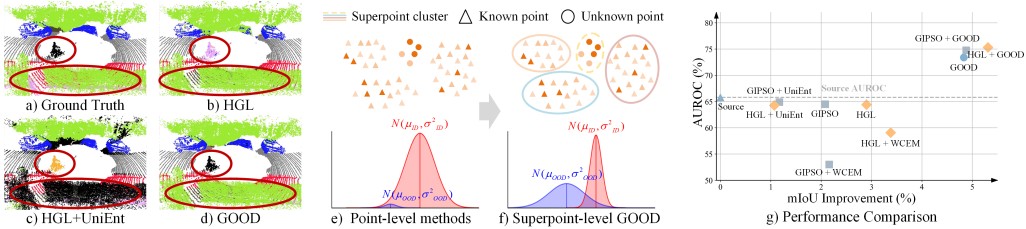

Figure 1: a) Black pixels indicate the OOD points in the target domain, such as the bicycle and bicyclist. b) The TTA-3DSeg model's (HGL (Zou et al., 2024)) overconfidence leads to erroneous predictions of dynamic OOD points as stationary vegetation, which could lead to serious safety incidents. c) Directly adapting the existing OSTTA method (UniEnt (Gao et al., 2024)) to point cloud semantic segmentation would lead to the misclassification of dominant ID points as OOD. d) Our GOOD effectively mitigates the imbalance between ID and OOD points, achieving satisfactory performance in both open-set and closed-set. e) Due to the predominance of ID points, point-level OSTTA methods would misclassify a large number of ID points as OOD, undermining segmentation performance. f) GOOD addresses this by grouping individual points into geometrically connected superpoints. Darker colors indicate higher uncertainty. g) We compare GOOD with GIPSO (Saltori et al., 2022), HGL (Zou et al., 2024), +*UniEnt* (Gao et al., 2024), +*WCEM* (Lee et al., 2023).

2022) and HGL (Zou et al., 2024), with 2D OSTTA approaches even results in reduced OOD detection performance compared to using these TTA methods alone. We speculate that this performance drop stems from the reliance of existing OSTTA methods on image instance-level processing; when transferred to 3D point clouds, such methods often fail to capture the geometric priors and spatial interdependencies essential for effective 3D discrimination, leading to limited OOD detection capabilities. Furthermore, these methods typically assume the availability of abundant OOD samples to assist in discerning discrepancies between in-distribution (ID) and OOD data. This assumption may not hold in 3D point cloud segmentation, where OOD samples could be very sparse.

In this paper, we delve into the open-set test-time adaptation for 3D point cloud semantic segmentation (OSTTA-3DSeg). As shown in Fig. 1 e) and f), rather than directly identifying imbalance individual points, we propose focusing on geometrically connected superpoints, which can better differentiate ID and OOD objects to relieve the imbalance in the quantity of ID and OOD points. To implement this approach, we develop a method termed Geometry-guided Out-of-Distribution (GOOD) modeling. Specifically, GOOD begins by clustering the point cloud into superpoints, effectively transforming the identification task from point-level to superpoint-level. Unlike existing OSTTA methods that purely rely on Gaussian Mixture Modeling (GMM) for pseudo-identification of ID and OOD data, we introduce a superpoint ID prototype strategy. This mechanism enhances the identification of ID superpoints, addressing false identifications by GMM, especially when OOD samples are sparse or absent. Furthermore, we incorporate temporal consistency to generate reliable pseudo-labels for stable and efficient online optimization. Similar to most existing TTA methods (Yuan et al., 2023; Wang et al., 2022), GOOD operates under an exponential moving average (EMA) framework. We validate GOOD across extensive experiments on four benchmarks with varying distribution-shift scenarios, showing that GOOD achieves new state-of-the-art performance.

Our contributions can be summarized as follows:

- To the best of our knowledge, we are the first to address open-set test-time adaptation for 3D point cloud semantic segmentation (OSTTA-3DSeg).
- We propose Geometry-guided Out-of-Distribution Modeling (GOOD), an effective method that shifts from point-level to superpoint-level identification, leading to more robust and coherent segmentation results.
- GOOD is designed to be flexibly integrated with existing TTA-3Dseg methods, substantially enhancing performance in open-set scenarios.

## 2  RELATED WORK

**Test-time Adaptation:** Test-time adaptation (TTA) has gained considerable attention due to its ability to adapt models to target data distributions without requiring access to source domain data during deployment. Extensive research has explored TTA in the image domain across tasks such as recognition (Wang et al., 2020; Niu et al., 2022; Chen et al., 2022), detection (Kim et al., 2022;

VS et al., 2023; Veksler, 2023), and segmentation (Wang et al., 2023; Zhang et al., 2022). Recently, there have been some efforts (Saltori et al., 2022; Zou et al., 2024; Weijler et al., 2024; Wang et al., 2024) made to TTA-3DSeg. GIPSO (Saltori et al., 2022) is the first method tailored for TTA-3DSeg, generating pseudo-labels for each category based on confidence-ranked proportions and employing additional networks for pseudo-label propagation. HGL (Zou et al., 2024) introduces a local-global pseudo-label strategy to balance pseudo-label accuracy with computational efficiency. Despite their effectiveness in handling covariate shifts, these methods largely overlook semantic shifts, where novel, unseen categories emerge in the target data. Addressing semantic shifts is critical, particularly in safety-critical applications such as autonomous driving, where robust recognition and response to novel objects are essential for reliable performance in the real-world.

**OOD Detection:** Traditional OOD detection usually enhances model reliability through advanced confidence metrics (Hendrycks & Gimpel, 2016; Hendrycks et al., 2019; Hsu et al., 2020; Liang et al., 2017; Liu et al., 2020) and strategically integrating auxiliary OOD data (Hendrycks et al., 2018; Katz-Samuels et al., 2022; Yang et al., 2021; Yu & Aizawa, 2019; Zhang et al., 2023). These approaches optimize the model's discriminative capability, enabling more precise differentiation between ID and OOD samples. More recently, several approaches (Li et al., 2023b; Lee et al., 2023; Gao et al., 2024; 2023; Yu et al., 2024; Schlachter & Yang, 2024) have emerged that optimize networks using unlabeled target data to enhance OOD detection at test time. While promising, these methods are generally limited to 2D images, which loses effectiveness in 3D point cloud segmentation, where the predominance of ID samples and the scarcity of OOD samples lead to frequent misclassification of ID points as OOD, compromising model robustness in open-world applications. Our work delves into this limitation and targets for proposing a dedicated framework for OSTTA-3DSeg, specifically designed to leverage geometric context for more robust segmentation in complex 3D environments.

## 3 METHODOLOGY

### 3.1 PROBLEM SETUP AND OVERVIEW

In this paper, we focus on open-set test-time adaptation for 3D point cloud semantic segmentation (OSTTA-3DSeg), which aims to adapt the source model $F_S$ online to the target model $F_T$ without access to the source data during model optimization. In this setting, the target domain $T$ and source domain $S$ exhibit both covariate shifts (distributional differences in shared classes) and semantic shifts (presence of new, unseen classes in the target domain). Formally, let $\mathcal{D}_S = \{(X_S^i, Y_S^i)\}_{i=1}^{N_S}$ represent the source domain dataset with label space $Y_S = \{1, ..., C_S\}$, and let $\mathcal{D}_T = \{(X_T^j, Y_T^j)\}_{j=1}^{N_T}$ represent the target domain dataset with label space $Y_T = \{1, ..., C_S, ..., C_T\}$, where $C_S$ and $C_T$ denote the number of classes in the source and target domains, respectively. In traditional closed-set TTA, $C_S = C_T$, whereas in OSTTA, $C_S < C_T$. The objective is to accurately segment points from the known (ID) classes $C_S$ while rejecting points from the unknown (OOD) classes in $C_T \setminus C_S$, even amidst substantial ID-OOD imbalance.

Traditional 2D OOD detection methods often treat individual instances in isolation, but for 3D data, spatial coherence and contextual relationships are crucial for accurate detection. To address this, we introduce the Geometry-guided Out-of-Distribution Modeling (GOOD) approach, as depicted in Fig. 2. GOOD utilizes a self-training strategy based on the mean-teacher EMA framework, where the teacher model $F_{T,tea}$ generates pseudo-labels that the student model $F_{T,stu}$ leverages for model optimization. To realize robust ID-OOD separation and accurate ID segmentation, GOOD incorporates two key branches within the teacher model: the Superpoint Representation Branch for effective detection of OOD points by grouping spatially coherent regions, and the Temporal Pseudo-label Branch to improve ID category identification by enforcing consistency across sequential frames. These components enable GOOD to capitalize on both temporal and geometric patterns involved in 3D data, enhancing OOD detection and ID segmentation performance.

### 3.2 SUPERPOINT REPRESENTATION BRANCH

In applications like autonomous driving, point cloud segmentation models often encounter far fewer unknown (OOD) points compared to known (ID) points, leading to significant class imbalance. This imbalance can undermine instance-based methods (Li et al., 2023b; Gao et al., 2024; Schlachter & Yang, 2024), resulting in frequent misclassifications. To address this challenge, we propose the

Figure 2: Method overview. a) GOOD consists of two branches: the Superpoint Representation Branch and the Temporal Pseudo-label Branch. The Superpoint Representation Branch further includes three key components—Superpoint Clustering, Superpoint Confidence, and Superpoint ID Prototypes. Specifically, b) Superpoint Clustering generates superpoints by aggregating multiple frames of point clouds and applying a clustering algorithm. Then, c) Superpoint Confidence introduces superpoint purity and superpoint entropy as metrics to distinguish between ID and OOD superpoint preliminarily. Finally, to address the noise and potential absence of OOD points, we additionally introduce d) Superpoint ID Prototypes to further refine the classification.

Superpoint Representation Branch (SRB), as illustrated in Fig. 2 b), c), and d). The SRB transforms discrete individual points into spatially correlated superpoints to mitigate extreme ID-OOD imbalance. Specifically, we first aggregate multiple frames of point clouds and apply a non-learning clustering method to generate superpoints, capturing spatial coherence across points. We then define superpoint purity and superpoint entropy to quantify confidence: purity measures the homogeneity of ID or OOD points within each superpoint, while entropy indicates confidence based on distributional uncertainty. Based on these metrics, a Gaussian Mixture Model (GMM) is then employed to differentiate ID and OOD superpoints. Finally, we establish superpoint ID prototypes to further refine the identification of ID superpoints, addressing potential misclassifications by GMM, particularly in cases where OOD samples are sparse or absent.

**Superpoint Clustering.** We propose a heuristic pre-segmentation to divide the point cloud into a set of superpoints, with each superpoint demonstrating high purity and typically containing points from only one category. Outdoor LiDAR scans, unlike indoor scenes, often feature well-separated objects, especially after ground points are identified and removed. Inspired by this philosophy, we design an intuitive approach to pre-segment each LiDAR frame, including three steps: 1) Combine temporal frames. Using the ego poses, we align the current and previous frames to produce a denser point cloud, which facilitates clustering by improving the continuity of object structures. 2) Remove ground points. To isolate objects from the ground, we apply RANSAC (Fischler & Bolles, 1981) to estimate the ground surface. To account for variations in ground slope, we partition the point cloud into smaller sub-regions and apply RANSAC to each sub-region individually. 3) Cluster into superpoints. We apply DBSCAN (Ester et al., 1996) to the remaining point cloud to form superpoints. Its ability to identify clusters of arbitrary shapes without a preset cluster number makes it ideal for complex outdoor LiDAR data and improves noise robustness.

**Superpoint Confidence.** Calculating prediction confidence and using this information for reliable OOD detection is essential in OSTTA. With the intuition that points within known (ID) superpoints exhibit greater geometric and temporal consistency than those within unknown (OOD) superpoints (confirmed in Fig. 2 c) right and Appendix A.3.5), we first introduce the metric of superpoint purity $\mathcal{C}_k^{pur}$ to evaluate the internal consistency of each superpoint. Formally, given a set of superpoints $S = \{s_k\}_{k=1}^K$ and the corresponding softmax predictions $P = \{\hat{p}_k\}_{k=1}^K$, we normalize the label $\hat{y}_{j,c}$ (the $c$-th element of the one-hot label for point $j$ in superpoint $s_k$) within each superpoint to obtain the probability distribution $\mathcal{P}_{k,c}$, and then calculate the superpoint purity $\mathcal{C}_k^{pur}$ as:

$$\{s_k, \hat{p}_k\}_{k=1}^K \leftarrow \text{DBSCAN}(\{x_i\}_{i=1}^N), \hat{p}_k \in R^{\mathcal{N}_k \times C},$$

$$\hat{y}_{j,c} = \mathbf{1}_c \left( \arg\max_c \{\hat{p}_{k,c}\}_{c \in C} \right), \mathcal{P}_{k,c} = \frac{1}{\mathcal{N}_k} \sum_{j \in \mathcal{N}_k} \hat{y}_{j,c},$$

$$\mathcal{C}_k^{pur} = 1 - \frac{1}{\log C} \sum_{c=1}^C \mathcal{P}_{k,c} \log \mathcal{P}_{k,c},$$

(1)

where $\mathbf{1}_c(.)$ denotes the one-hot transformation, $\mathcal{N}_k$ is the point number in superpoint, $C$ is the number of known classes, and $\hat{p}_{k,c}$ denotes the softmax probability of point in superpoint $s_k$ belonging

to the $c$-th class. However, superpoint purity alone may be insufficient in cases where the majority of points show only a slight preference for one class over others, making the metric less reliable. To address this, we further introduce the superpoint entropy $\mathcal{C}_k^{ent}$, a soft version of Eq. 1 that replaces the $\mathbf{1}_c(\arg\max(.))$ term with $\mathrm{softmax}(.)$ to better capture distributional uncertainty.

Accordingly, we obtain the final function of superpoint confidence as $\mathcal{C}_k^{sup} = \mathcal{C}_k^{ent} \cdot \mathcal{C}_k^{pur}$, which is empirically validated in Appendix A.3.5 to achieve a good trade-off between closed-set and open-set performance. As shown in Fig. 2 c), we empirically observe that the bimodal distribution of $\mathcal{C}_k^{sup}$ effectively distinguishes ID and OOD superpoints, with two distinct peaks corresponding to each type. Consequently, we model the distribution of $\mathcal{C}_k^{sup}$ using a two-component GMM, i.e. $\mathcal{G}(x)$. Here, the component with the larger mean represents the ID superpoints, while the component with the smaller mean corresponds to the OOD superpoints.

$$\mathcal{G}(x) = \pi(x)\mathcal{N}\left(x \mid \mu_{\mathrm{ID}}, \sigma_{\mathrm{ID}}^2\right) + (1 - \pi(x))\mathcal{N}\left(x \mid \mu_{\mathrm{OOD}}, \sigma_{\mathrm{OOD}}^2\right), \tag{2}$$

where $\pi(x)$ denotes the probability that $\mathcal{C}_k^{sup}$ belongs to the ID superpoint, $\mu_{\mathrm{ID}}, \sigma_{\mathrm{ID}}^2$ and $\mu_{\mathrm{OOD}}, \sigma_{\mathrm{OOD}}^2$ denote the mean and variance of the ID and OOD superpoint, respectively. Although superpoints alleviate the extreme ID-OOD imbalance, ID superpoints may still be misclassified as OOD. To address this, we discard mixed data in the middle, classifying superpoints below threshold $\mu_{\mathrm{OOD}}$ as OOD $S_{OOD}$ and above threshold $\mu_{\mathrm{ID}}$ as ID $S_{ID}$.

**Superpoint ID Prototypes.** While the GMM based on superpoint confidence enhances ID and OOD discrimination, some limitations remain. Specifically, superpoint confidence relies solely on the classifier's logit output, overlooking embedding features of the superpoints. This omission can lead to mixed ID and OOD superpoints. Furthermore, in certain scenarios, OOD samples may be absent. Nevertheless, the GMM-based partitioning method forces each point cloud frame to contain both ID and OOD superpoints, potentially resulting in the over-segmentation of OOD regions and degrading model performance. Moreover, we observe that while OOD superpoints may be inconsistently identified, ID superpoints tend to be more stable and reliable, as shown in Fig. 2 c) right. These observations motivate our development of a superpoint ID prototype strategy to achieve more accurate and robust identification.

Concretely, at each timestep $t$, we construct a set of prototypes for each ID class based on superpoints $S_{ID}$. For each class $c$, we calculate the prototype $\rho_c^t$ as the centroid of all ID superpoint embeddings, defined as $\rho_c^t = \frac{1}{\mathcal{N}_c}\sum_k^{\mathcal{N}_c} z_k^t$, where $\mathcal{N}_c$ is the number of superpoint of current class $c$ and $z_k^t$ is the mean embedding feature of each superpoint. To stabilize training, we update the prototypes incrementally using EMA: $\hat{\rho}_c^t = \alpha\hat{\rho}_c^{t-1} + (1-\alpha)\rho_c^t$. Empirically, we set EMA coefficient $\alpha = 0.99$. Then we can evaluate the similarity between pseudo-labeled OOD superpoints and ID prototypes, identifying potential misclassifications among the OOD superpoints. This is formalized as:

$$\hat{\mathbf{s}}_k = \begin{cases} \mathrm{OOD}, & \text{if } \mathrm{sim}\left(z_k^t, \hat{\rho}_c^t\right) < \tau \\ \mathrm{ID}, & \text{if } \mathrm{sim}\left(z_k^t, \hat{\rho}_c^t\right) \geq \tau \end{cases}, \tag{3}$$

where $\mathrm{sim}(a, b)$ measures the cosine similarity between $a$ and $b$, $\tau$ is an empirical threshold. We define $\hat{Y}_{OOD} = \sum_{k \in OOD} \hat{\mathbf{s}}_k$ as OOD pseudo-label.

In summary, the dedicated SRB reliably separates ID and OOD superpoints, effectively addressing extreme cases where OOD samples are minimal or absent.

### 3.3 TEMPORAL PSEUDO-LABEL BRANCH

Existing TTA-3DSeg methods (Saltori et al., 2022; Zou et al., 2024) typically assign pseudo-labels for ID classes based solely on single-frame information, overlooking valuable context provided by consecutive frames. To overcome this limitation, we propose an effective Temporal Pseudo-labels Branch (TPB) that fully leverages temporal information. Technically, we start by inputting the current point cloud $X^t$ into the teacher model $F_{T,tea}^t$, generating the softmax prediction $P^t$ and one-hot label $\hat{Y}^t$. Subsequently, we project both the current frame $(X^t, P^t)$ and the previous frames $(X^{t-1}, P^{t-1}), ..., (X^{t-w}, P^{t-w})$ into the global coordinate system. Following prior work (Saltori et al., 2022; Zou et al., 2024), we assume access to reliable self-pose for temporal alignment; if this assumption is violated (i.e., the pose is noisy), performance may degrade without targeted adjustments. Further analyses of noise effects, alternative solutions, limitations of this assumption, and verification details are provided in the Appendix A.3.6. For each point $x_i^t$ in the $X^t$, we then leverage K-NN to identify temporal neighborhoods $Ne_i^{t-1}, ..., Ne_i^{t-w}$ from each preceding frame.

Table 1: Results of different methods on Synth4D to SemanticKITTI. ↑ indicates that larger values are better, and vice versa.

| Model | Vehicle | Pedestrian | Road | Sidewalk | Terrain | Manmade | Vegetation | mIoU(↑) | AUROC(↑) | FPR95(↓) |
|---|---|---|---|---|---|---|---|---|---|---|
| Source | 60.95 | 15.58 | 73.57 | 27.14 | 16.36 | 32.04 | 56.20 | 40.26 | 65.80 | 78.39 |
| BN (Nado et al., 2020) | -1.47 | -0.82 | -4.35 | -5.13 | +6.55 | +10.03 | +2.15 | +0.99 | 65.36 | 79.98 |
| TENT (Wang et al., 2020) | -1.28 | -0.68 | -4.41 | -5.19 | +6.61 | +10.11 | +2.41 | +1.08 | 65.26 | 83.89 |
| ConjugatePL (Goyal et al., 2022) | -0.87 | -0.23 | -4.72 | -5.40 | +6.76 | +10.36 | +3.28 | +1.30 | 65.96 | 79.91 |
| GIPSO (Saltori et al., 2022) | +10.57 | -6.12 | +0.86 | -5.26 | +8.36 | +3.31 | +2.82 | +2.08 | 64.47 | 80.40 |
| +*SeaT* | +9.39 | -6.54 | -0.54 | -5.64 | +8.84 | +3.96 | +3.21 | +1.81 | 54.46 | 95.41 |
| +*UniEnt* (Gao et al., 2024) | +7.53 | -7.12 | -1.37 | -5.50 | +8.94 | +3.30 | +2.46 | +1.17 | 64.90 | 82.27 |
| +*WCEM* (Li et al., 2023b) | +11.26 | -6.35 | -1.43 | -7.31 | +10.09 | +4.58 | +4.30 | +2.16 | 53.01 | 100.0 |
| +*GOOD* | +12.13 | -2.15 | +1.64 | -1.29 | +7.98 | +10.96 | +4.94 | **+4.88** | **74.76** | **73.30** |
| HGL (Zou et al., 2024) | +10.13 | -7.33 | -0.10 | +0.37 | +8.60 | +5.97 | +2.67 | +2.90 | 64.40 | 79.82 |
| +*SeaT* | +4.87 | -7.93 | -3.29 | +0.30 | +7.48 | +6.36 | +2.24 | +1.43 | 66.43 | 75.99 |
| +*UniEnt* (Gao et al., 2024) | +6.45 | -6.57 | -6.73 | -1.08 | +8.56 | +5.75 | +1.15 | +1.07 | 64.28 | 83.34 |
| +*WCEM* (Li et al., 2023b) | +12.24 | -5.78 | -0.19 | -0.13 | +8.26 | +6.88 | +2.43 | +3.38 | 59.07 | 100.0 |
| +*GOOD* | +13.65 | -0.02 | +1.92 | -0.04 | +6.23 | +9.93 | +5.56 | **+5.31** | **75.31** | **71.31** |
| GOOD | +12.95 | +0.01 | +1.14 | -0.66 | +5.65 | +9.90 | +4.88 | **+4.83** | **73.39** | **71.91** |

The neighborhood label $y_i^{t,Ne}$ for $x_i^t$ is then calculated as the aggregated label from its neighboring points in previous frames:

$$\hat{y}_i^{t,Ne} = \mathbf{1}_c(\arg\max_c\{ \sum_{i \in Ne_i^{t-1},...,Ne_i^{t-w}} p_i\}_{c \in C}). \tag{4}$$

Next, we assign ID pseudo-labels $\hat{Y}_{ID}$ only to points where the predictions are consistent between the current pseudo-label and its temporal neighborhood pseudo-label:

$$\hat{y}_{i,ID}^t = y_i^{t,Ne}, \text{s.t. } \hat{y}_i^t = \hat{y}_i^{t,Ne}. \tag{5}$$

Finally, we remove overlapping regions of $\hat{Y}_{OOD}$ from $\hat{Y}_{ID}$, resulting in a filtered ID pseudo-label.

Notably, the EMA-based teacher-student architecture in TPB not only explicitly enforces temporal consistency within the point cloud data but also implicitly promotes temporal stability within the model itself, thereby enhancing both accuracy and robustness. Additionally, it does not require extra threshold hyperparameters to balance precision and generalization.

### 3.4 TEMPORAL-BASED FEATURE REGULARIZATION

Following (Chen & He, 2021; Saltori et al., 2022; Zou et al., 2024), we also introduce temporal regularization to enforce consistency of projected features between corresponding points across consecutive frames:

$$\mathcal{D}_{t \to t-w}\left(q^t, z^{t-w}\right) = -\frac{q^t}{\|q^t\|_2} \cdot \frac{z^{t-w}}{\|z^{t-w}\|_2}, \mathcal{L}_{reg} = \frac{1}{2}\mathcal{D}_{t \to t-w}\left(q^t, z^{t-w}\right) + \frac{1}{2}\mathcal{D}_{t-w \to t}\left(q^{t-w}, z^t\right),$$
$$\tag{6}$$

where $q^t$ and $z^{t-w}$ are the projected feature of corresponding points in frames $t$ and $t-w$, generated by an additional encoder network $h(\cdot)$ and predictor head $f(\cdot)$, respectively. The temporal consistency loss $\mathcal{L}_{reg}$ is defined as the average bidirectional feature distance between frames. Please refer to Appendix A.2.1 for more details.

### 3.5 TEST-TIME MODEL ADAPTATION

Given the class imbalance in point clouds and to prevent confusion between unknown and known points, we adopt the Dice loss $\mathcal{L}_{dice}$ as the ID pseudo-label learning objective. Additionally, we introduce an entropy-based OOD loss, $\mathcal{L}_{ood} = -\frac{1}{\|S_{OOD}\|}\sum_{x \in S_{OOD}} H\left(F_{T,stu}(x)\right)$ to supervise pseudo-labeled OOD superpoints, where $H(.)$ denotes the entropy. The overall test-time optimization objective for the student model is defined as $\mathcal{L}_{final} = \mathcal{L}_{dice} + \mathcal{L}_{reg} + \lambda\mathcal{L}_{ood}$, where $\lambda$ is a trade-off hyperparameter. During the iterative training process, the teacher model is updated indirectly by the student model using the EMA strategy.

## 4 EXPERIMENTS

### 4.1 SETUP

**Source and Target Datasets:** Following previous studies, we evaluate our method on the widely used Lidar benchmark datasets: **Synth4D** (Saltori et al., 2022), **SynLiDAR** (Xiao et al., 2021), **SemanticKITTI** (Behley et al., 2019) and **nuScenes** (Caesar et al., 2020). The source model is

Table 2: Results of different methods on Synth4D to nuScenes.

| Model | Vehicle | Pedestrian | Road | Sidewalk | Terrain | Manmade | Vegetation | mIoU(↑) | AUROC(↑) | FPR95(↓) |
|---|---|---|---|---|---|---|---|---|---|---|
| Source | 24.69 | 17.29 | 68.66 | 17.68 | 9.34 | 58.89 | 47.75 | 35.59 | 60.44 | 76.97 |
| BN (Nado et al., 2020) | +0.45 | -0.44 | -0.17 | +0.01 | +0.71 | -0.40 | +1.27 | +0.31 | 60.66 | 76.85 |
| TENT (Wang et al., 2020) | +0.34 | -0.37 | +0.42 | +0.17 | +0.25 | +0.38 | +0.61 | +0.25 | 60.57 | 76.88 |
| ConjugatePL (Goyal et al., 2022) | +0.87 | -0.41 | +0.62 | +0.69 | +0.50 | +0.56 | +1.06 | +0.64 | 61.11 | 76.71 |
| GIPSO (Saltori et al., 2022) | +1.89 | -1.63 | -4.81 | +1.12 | +3.74 | -0.35 | +3.59 | +0.48 | 58.66 | 78.75 |
| +SeaT | +0.91 | -1.82 | -6.56 | +0.83 | +2.71 | -1.76 | +1.77 | -0.60 | 55.97 | 81.40 |
| +UniEnt (Gao et al., 2024) | +1.55 | -1.86 | -6.19 | +0.62 | +3.53 | -1.00 | +3.32 | -0.04 | 58.77 | 77.65 |
| +WCEM (Li et al., 2023b) | +2.48 | -1.24 | -0.92 | +1.40 | +4.12 | -0.91 | +2.25 | +1.01 | 57.80 | 83.51 |
| +GOOD | -0.15 | -0.37 | +0.94 | +1.47 | +3.69 | +0.85 | +5.23 | +1.67 | 62.95 | 73.25 |
| HGL (Zou et al., 2024) | +2.49 | -4.21 | -2.82 | -0.18 | +5.07 | -0.13 | +3.40 | +0.51 | 57.67 | 82.01 |
| +SeaT | +1.61 | -4.03 | -2.04 | -0.45 | +3.97 | -1.17 | +1.07 | -0.15 | 57.75 | 79.15 |
| +UniEnt (Gao et al., 2024) | +1.56 | -4.63 | -4.36 | -0.74 | +4.84 | -1.09 | +2.44 | -0.30 | 57.42 | 81.02 |
| +WCEM (Li et al., 2023b) | +2.63 | -3.39 | -1.69 | -0.11 | +5.30 | -0.39 | +2.43 | +0.66 | 57.14 | 84.77 |
| +GOOD | +1.04 | +0.11 | +3.67 | +1.88 | +3.53 | +2.07 | +5.75 | +2.62 | 62.20 | 74.82 |
| GOOD | +0.51 | -0.21 | +3.08 | +1.79 | +3.15 | +1.53 | +5.53 | +2.23 | 64.60 | 72.60 |

Table 3: Results of different methods on SynLiDAR to SemanticKITTI.

| Model | car | bi.cle | mt.cle | truck | pers. | b.clst | m.clst | road | park. | sidew. | oth-g. | build. | fence | veget. | trunk | terra. | pole | traff. | mIoU(↑) | AUROC(↑) | FPR95(↓) |
|---|---|---|---|---|---|---|---|---|---|---|---|---|---|---|---|---|---|---|---|---|---|
| Source | 56.13 | 3.82 | 17.86 | 16.77 | 17.49 | 16.49 | 33.32 | 73.14 | 10.53 | 40.30 | 0.02 | 39.07 | 8.67 | 63.90 | 15.73 | 28.50 | 32.45 | 10.89 | 26.95 | 54.90 | 86.67 |
| BN (Nado et al., 2020) | +4.37 | +1.04 | +1.76 | +7.04 | +1.12 | -0.07 | +2.72 | +1.21 | -1.89 | -0.04 | -0.01 | +4.65 | -0.46 | -1.04 | -0.53 | +0.28 | -0.70 | +0.35 | +1.10 | 55.60 | 85.29 |
| TENT (Wang et al., 2020) | +6.52 | +1.32 | +2.83 | +9.50 | +2.62 | +1.33 | +2.17 | -1.39 | -3.87 | -3.23 | -0.01 | +11.78 | -0.90 | -2.59 | +0.66 | -3.81 | -1.92 | +1.39 | +1.19 | 58.27 | 81.15 |
| ConjugatePL (Goyal et al., 2022) | +6.15 | +1.05 | +2.66 | +9.74 | +2.68 | +4.19 | -1.20 | -1.86 | -3.17 | -3.93 | +0.00 | +12.51 | -0.32 | -0.02 | +3.62 | -5.59 | -3.03 | +2.30 | +1.38 | 60.11 | 80.09 |
| GIPSO (Saltori et al., 2022) | +4.04 | -0.23 | +1.14 | +2.56 | -2.26 | -5.26 | -4.81 | +0.58 | +5.64 | +3.24 | +0.02 | +3.08 | +1.70 | +1.46 | +4.11 | +4.90 | +2.79 | +3.66 | +1.46 | 57.95 | 82.97 |
| +SeaT | +6.54 | -0.92 | -0.94 | +5.23 | -3.23 | -6.77 | -3.57 | -2.50 | +5.10 | +1.98 | +0.05 | +3.05 | +2.13 | +1.21 | +4.18 | +4.32 | +1.21 | +3.69 | +1.15 | 56.28 | 85.30 |
| +UniEnt (Gao et al., 2024) | +13.10 | -0.92 | -0.55 | +8.77 | -3.80 | -7.80 | -0.49 | -0.95 | +6.07 | +2.47 | +0.18 | +6.00 | +2.34 | -10.29 | +4.07 | -0.68 | +2.50 | +3.01 | +1.28 | 44.13 | 85.95 |
| +WCEM (Li et al., 2023b) | +4.29 | -0.25 | +1.33 | +2.18 | -2.63 | -4.62 | -0.43 | +1.07 | +5.80 | +3.75 | +0.02 | +3.81 | +1.60 | +1.54 | +3.94 | +5.37 | +3.60 | +4.27 | +1.92 | 62.57 | 100.0 |
| +GOOD | +17.16 | +0.82 | +4.39 | +16.04 | -0.13 | -3.35 | +6.00 | +6.65 | +4.19 | +3.21 | -0.02 | +16.11 | +1.13 | +6.45 | +4.58 | -0.42 | +5.92 | +6.16 | +5.27 | 71.02 | 71.66 |
| HGL (Zou et al., 2024) | +10.99 | -0.40 | -4.74 | +7.55 | -2.99 | -3.34 | -18.80 | +1.43 | +5.13 | +3.49 | +0.01 | +13.26 | +2.90 | +2.89 | +6.04 | +4.58 | +3.75 | +2.54 | +1.90 | 62.91 | 79.13 |
| +SeaT | +9.88 | -1.14 | -2.85 | +7.05 | -3.52 | -5.81 | -8.52 | -2.31 | +4.87 | +3.72 | +0.10 | +10.46 | +2.95 | +4.45 | +4.94 | +4.46 | +5.39 | +4.14 | +1.68 | 60.70 | 79.62 |
| +UniEnt (Gao et al., 2024) | +11.26 | -1.68 | -1.91 | +10.61 | -4.50 | -3.80 | -5.08 | -2.13 | +4.50 | +3.80 | +0.03 | +11.32 | -1.63 | -3.76 | +2.73 | -1.63 | -3.76 | +2.73 | +1.31 | 56.08 | 100.0 |
| +WCEM (Li et al., 2023b) | +13.47 | -0.21 | -5.48 | +8.60 | -2.54 | -6.55 | -15.71 | +0.93 | +4.87 | +4.12 | +0.03 | +11.71 | +2.90 | +4.41 | +6.16 | +6.17 | +5.14 | +3.14 | +2.28 | 71.05 | 100.0 |
| +GOOD | +17.07 | +0.70 | +2.93 | +13.14 | +0.16 | -2.58 | +4.10 | +6.59 | +4.49 | +3.81 | +0.00 | +15.33 | +1.60 | +6.50 | +3.95 | -0.32 | +5.72 | +3.88 | +4.83 | 72.02 | 68.94 |
| GOOD | +17.67 | -0.80 | +3.39 | +14.85 | +1.59 | -2.61 | +6.47 | +5.99 | +1.72 | +3.18 | -0.01 | +17.00 | +1.40 | +6.58 | +2.96 | -1.31 | +7.15 | +6.04 | +5.03 | 73.23 | 69.96 |

pre-trained on virtual datasets (Synth4D or SynLiDAR) and adapted to real-world datasets (SemanticKITTI or nuScenes). See Appendix A.3.1 for more details.

**Label Mapping:** Since no standard setup currently exists for OSTTA-3DSeg, we propose standardized training and evaluation settings to facilitate future research in this area. We design virtual-to-real experiments based on LiDAR beam configurations as: Synth4D-64 to SemanticKITTI, Synth4D-32 to nuScenes, and SynLiDAR to SemanticKITTI. For the first two experiments, following GIPSO (Saltori et al., 2022), we remap the SemanticKITTI and nuScenes datasets to match Synth4D by selecting seven common classes and treating all remaining categories as OOD classes. In the SynLiDAR to SemanticKITTI experiment, we follow (Cen et al., 2022), designating *other-vehicle* as the OOD class, while all other categories are treated as ID classes.

**Evaluation Protocol:** Following recent researches (Saltori et al., 2022; Zou et al., 2024), we evaluate the performance of TTA methods on a new incoming frame utilizing the model that has been adapted to the preceding frame. To evaluate the adapted model in an open-world setting, we consider both closed-set and open-set metrics. Specifically, the closed-set performance is measured by the improvement in mean intersection-over-union (mIoU) over the source model, while open-set performance is evaluated using the area under the receiver operating characteristic curve (AUROC) and the false positive rate of OOD samples at a 95% true positive rate for ID samples (FPR95).

**Baseline Methods:** We mainly compare our method with TTA-3DSeg baselines, such as **GIPSO** (Saltori et al., 2022) and **HGL** (Zou et al., 2024), and image-based OSTTA methods, including **UniEnt** (Gao et al., 2024) and **WCEM** (Lee et al., 2023). In addition, we introduce a straightforward baseline based on hyperparameter thresholds searching to distinguish ID and OOD points, termed Searching Thresholds (**SeaT**). This strategy identifies OOD points by selecting points with the lowest confidence.

**Implementation Details:** Following (Saltori et al., 2022; Zou et al., 2024), we use the MinkowskiUNet-18 (Choy et al., 2019) as our backbone and set the voxel size to 0.05. For online model adaptation, the batch size is set to 1 across all benchmark datasets. We configure hyperparameters as $w = 3$, $\tau = 0.85$, and $\lambda = 0.1$ for all datasets. For temporal K-NN, K is set to 10 for SemanticKITTI and 5 for nuScenes. Further details on DBSCAN configuration and additional implementation information can be found in the Appendix A.3.5.

## 4.2 EXPERIMENT RESULTS

**Synth4D to SemanticKITTI and nuScenes:** We first evaluate our method on the Synth4D to SemanticKITTI and Synth4D to nuScenes tasks. As shown in Tables 1 and 2, GOOD significantly enhances both closed-set ID segmentation and open-set OOD identification performance. Specifically,

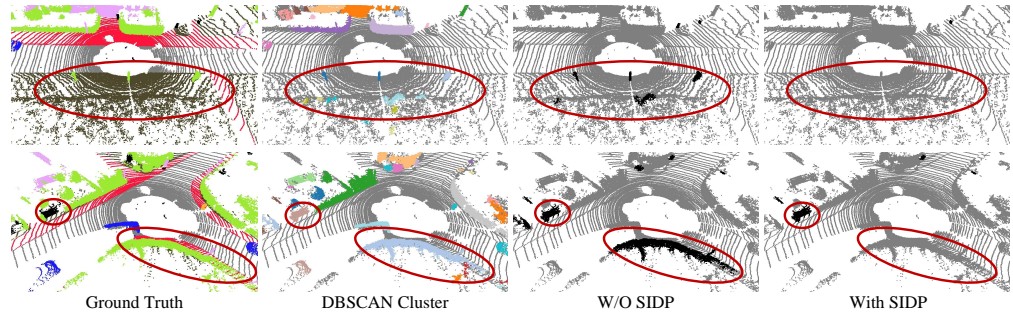

Ground Truth      DBSCAN Cluster      W/O SIDP      With SIDP

Figure 3: Qualitative results of Superpoint ID Prototypes (SIDP) on Synth4D to SemanticKITTI. Black pixels represent the open-set points in ground truth or unknown pseudo-label. The results indicate that SIDP effectively corrects ID misclassifications and handles absent OOD.

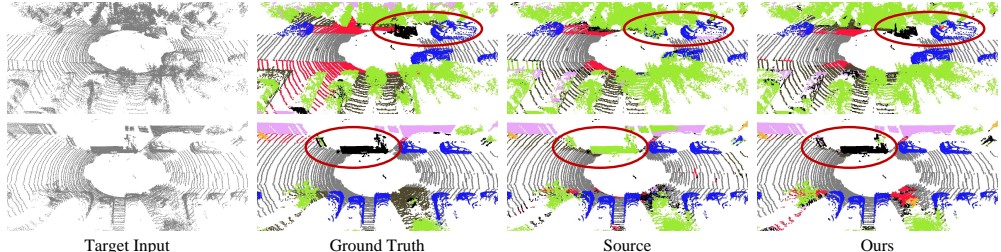

Target Input      Ground Truth      Source      Ours

Figure 4: Qualitative results on Synth4D to SemanticKITTI. Black pixels denote open-set points in the ground truth and prediction OOD points with Maximum Softmax Probability (MSP) above 0.6. Our method significantly enhances both ID and OOD segmentation.

on Synth4D to SemanticKITTI, GOOD achieves improvements in mIoU, AUROC, and FPR95 of +4.83%, 73.39%, and 71.91%, respectively, marking gains of 1.93%, 8.99%, and 7.91% over HGL. Similarly, on Synth4D to nuScenes, GOOD yields mIoU, AUROC, and FPR95 values of +2.23%, 64.60%, and 72.60%, with respective improvements of 1.72%, 6.93%, and 9.41% over HGL. Furthermore, GOOD can complement existing methods effectively. For example, integrating GOOD with HGL on Synth4D to SemanticKITTI enhances mIoU from +2.90% to +5.31%, AUROC from 64.40% to 73.39%, and reduces FPR95 from 79.82% to 71.91%.

In comparison, the 2D image method +*UniEnt* (Gao et al., 2024) negatively impacts closed-set 3D segmentation mIoU and has limited effect on open-set AUROC and FPR95. This is primarily due to its inability to effectively differentiate between imbalanced ID and OOD points, resulting in the erroneous application of entropy maximization to many ID points and leading to suboptimal results. The addition of +*WCEM* (Li et al., 2023b) yields a noticeable improvement in closed-set performance but performs poorly on open-set metrics. This is also due to its difficulty in distinguishing ID from OOD points, which causes the loss function to degenerate into entropy minimization.

**SynLiDAR to SemanticKITTI:** We then conduct experiments on SynLiDAR to SemanticKITTI. In this setup, the imbalance between known and unknown points is exacerbated, with many frames lacking any unknown points. As shown in Table 3, consistent with our previous findings, GOOD achieves superior performance. Specifically, GOOD yields mIoU improvement, AUROC, and FPR95 scores of +5.03%, 73.23%, and 69.96%, marking gains of 3.13%, 10.32%, and 9.17% over HGL, respectively. Additionally, the increased imbalance between ID and OOD points causes a notable decline in both closed-set and open-set performance for the +*SeaT* and +*UniEnt* methods.

**Efficiency:** The last column of Table 4 presents the adaptation times. Our method outperforms GIPSO in speed but is slightly less efficient than HGL, primarily due to the multi-frame superpoint clustering step, which takes about 0.7 seconds. Notably, under closed-set conditions without the Superpoint Representation Branch (SRB), our method achieves a speed comparable to HGL while outperforming it by 1.71% in mIoU, as shown in Experiment I. More experiments and discussions regarding clustering algorithms and runtime can be found in the Appendix A.3.4 and A.3.5.

**Qualitative Results:** Fig. 4 shows the comparison between GOOD and the source model on Synth4D to SemanticKITTI, demonstrating that GOOD consistently improves segmentation performance for both ID and OOD points.

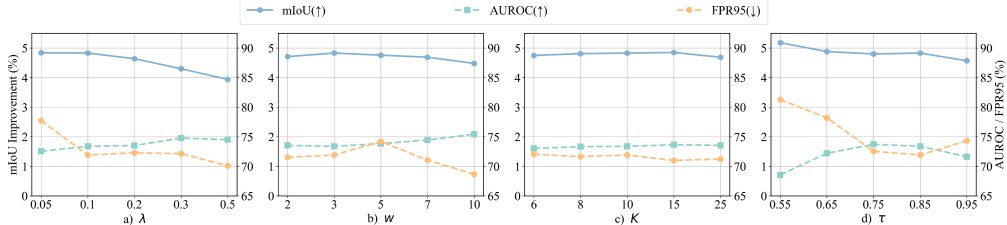

Figure 5: Hyper-parameter sensitivity analysis on Synth4D to SemanticKITTI.

Table 4: Ablation Study. TP, SC, and SIDP denote Temporal Pseudo-labels, Superpoint Confidence and Superpoint ID Prototypes, respectively.

| ID | TP | SC | SIDP | mIoU(%) | AUROC(%) | FPR95(%) | Runtimes(s) |
|---|---|---|---|---|---|---|---|
| Source | - | - | - | 40.26 | 65.80 | 78.39 | - |
| GIPSO (Saltori et al., 2022) | - | - | - | +2.08 | 64.47 | 80.40 | 4.03 |
| HGL (Zou et al., 2024) | - | - | - | +2.90 | 64.40 | 79.82 | 0.65 |
| I | ✓ | - | - | +4.61 | 66.66 | 79.72 | 0.70 |
| II | ✓ | ✓ | - | +4.54 | 71.47 | 76.82 | 1.42 |
| III | ✓ | ✓ | ✓ | +4.83 | 73.39 | 71.91 | 1.45 |

Table 5: Results under different sparsity levels and OOD class counts. Values are presented as (mIoU(↑), AUROC(↑), FPR95(↓)).

| Unknown Setting | Source | Ours (w/o SRB) | Ours |
|---|---|---|---|
| w/o other-vehicle | (26.95, 54.90, 86.67) | (+4.77, 65.35, 77.86) | (+5.03, 73.23, 69.96) |
| w/o trunk | (26.31, 51.72, 91.56) | (+4.70, 54.42, 90.18) | (+4.68, 59.18, 87.51) |
| w/o car | (24.93, 66.23, 70.18) | (+3.33, 68.29, 74.88) | (+3.60, 71.90, 65.75) |
| w/o building | (24.87, 49.09, 89.12) | (+3.35, 55.21, 82.15) | (+3.12, 61.32, 75.62) |
| ≤ person | (30.87, 69.89, 75.41) | (+6.39, 71.50, 83.17) | (+6.27, 73.75, 75.26) |
| ≤ other-vehicle | (38.12, 65.10, 83.79) | (+4.26, 71.52, 72.39) | (+4.11, 75.87, 70.13) |
| ≤ fence | (50.46, 55.94, 92.41) | (+7.31, 64.30, 82.69) | (+7.44, 69.44, 82.56) |
| ≤ car | (46.48, 64.02, 90.47) | (+2.59, 65.99, 82.08) | (+2.68, 74.65, 76.73) |

## 4.3 ABLATION STUDY

We conduct an extensive ablation study to assess the contribution of each component. Unless otherwise specified, this study is conducted on Synth4D to SemanticKITTI.

**Component Analysis:** As shown in Tab. 4, we evaluate the effectiveness of the Temporal Pseudo-labels (TP), Superpoint Confidence (SC), and Superpoint ID Prototypes (SIDP). To assess the effectiveness of the proposed TP, we conduct experiments using only TP, and the result from Experiment I indicates that the standalone TP model significantly outperforms the baseline methods. Experiments I and II demonstrate that introducing SC significantly enhances open-set performance, despite a minor decline in closed-set performance. The results of Experiments II and III demonstrate the critical role of the SIDP, as it effectively reduces the misclassification of known superpoints, while simultaneously enhancing both open-set and closed-set performance. The visualization results in Fig. 3 also qualitatively demonstrate the effectiveness of SIDP.

**Hyper-parameter Sensitivity:** We first examine the sensitivity of $\lambda$ in Fig. 5 (a), where $\lambda$ takes values from $\{0.05, 0.1, 0.2, 0.3, 0.5\}$. The results indicate that larger $\lambda$ values improve open-set performance but may slightly reduce closed-set performance. For all benchmarks, we set $\lambda = 0.1$. We then explore the effect of $w$ in Fig. 5 (b), with $w$ ranging from $\{2, 3, 5, 7, 10\}$. The results show that our method is robust around the selected parameter $w = 3$. We then evaluate the hyperparameter $K$ in K-NN, with values ranging from 0 to 25. As shown in Fig. 5 (c), $K$ demonstrates stability around a value of 10. Finally, we assess $\tau$ in Fig. 5 (d). A smaller $\tau$ value sets a stricter criterion for identifying unknown superpoints, enhancing closed-set performance while reducing open-set performance, and vice versa. For all benchmarks, we predefine $\tau = 0.85$ to distinguish between ID and OOD superpoints.

**Performance under Different Sparsity levels and OOD Class Counts:** The sparsity level and OOD class counts represent the complexity of the open world. We examine the impact of different sparsity levels and unknown class counts in Tab. 5. For different sparsity levels, we perform experiments on the SynLiDAR to SemanticKITTI and control the data ratio at 0.46%, 1.15%, 6.47%, and 11.90%, designating *other-vehicle*, *trunk*, *car*, and *building* as an unknown class, respectively. For different class counts, we sort the class proportions from smallest to largest, discarding all classes with proportions less than or equal to *person*, *other-vehicle*, *fence*, and *car*. The proportions of unknown classes are 0.46%, 1.30%, 6.44%, and 12.92%, corresponding to unknown class counts of 7, 10, 13, and 14, respectively. As shown in Tab. 5, our GOOD and SRB can effectively address varying sparsity levels and OOD class counts.

Further ablation studies and additional experimental results are provided in the Appendix A.3.5.

## 5 CONCLUSION

In this paper, we introduce the *GOOD* framework, a *simple yet effective* approach for open-set test-time adaptation in point cloud semantic segmentation. Existing methods, primarily developed for 2D

image recognition, face significant challenges when applied to point cloud segmentation due to the complexities of point cloud data, especially the imbalance between ID and OOD points. To address these issues, our framework leverages the unique characteristics of point clouds by using superpoints to mitigate data imbalance. Technically, we devise the Superpoint Clustering method to organize the disordered point cloud, then propose the Superpoint Confidence metric to preliminarily distinguish ID and OOD superpoints, and finally introduce the Superpoint ID Prototypes to further refine the identification of ambiguous OOD superpoints. Furthermore, we propose a temporal-based pseudo-label generation method that simultaneously leverages both data temporal information and model temporal consistency. Extensive experiments demonstrate that GOOD significantly outperforms existing TTA-3DSeg methods in open-set scenarios. Notably, on the Synth4D to SemanticKITTI task, GOOD surpasses HGL by 8.99% and 7.91% in AUROC and FPR95, respectively.

**Acknowledgments**: This work was supported by the National Natural Science Foundation of China (No. 62372329), in part by the National Key Research and Development Program of China (No. 2024YFE0211000), in part by the National Natural Science Foundation of China (No. 62506263, 62506264), in part by the Shanghai Scientific Innovation Foundation (No. 23DZ1203400), in part by the Fundamental Research Funds for the Central Universities, in part by the Key Technology Development and Integrated Application of Guided Autonomous Vehicles Project, in part by the China Postdoctoral Science Foundation (No. BX20250383, GZB20250385, 2025M771530, 2025M771539), in part by Tongji-Qomolo Autonomous Driving Commercial Vehicle Joint Lab Project, and in part by Xiaomi Young Talents Program.

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

# A APPENDIX

## A.1 MORE RELATED WORK

**Point Cloud Semantic Segmentation:** Point cloud semantic segmentation is a fundamental computer vision task that assigns semantic categories to each point within a point cloud. Depending on the specific methodology used, approaches to point cloud segmentation can be broadly categorized into point-based methods (Qi et al., 2017a;b; Hu et al., 2020), range-map-based methods (Wu et al., 2018; Cortinhal et al., 2020; Milioto et al., 2019), sparse voxel-based methods (Choy et al., 2019;

Graham et al., 2018; Graham & Van der Maaten, 2017; Zhou & Tuzel, 2018), and other specialized approaches such as hybrid point-voxel (Tang et al., 2020), polar coordinate (Zhang et al., 2020), and cylindrical coordinate systems (Zhu et al., 2021). Although these methods achieve strong performance under closed-set conditions, they remain sensitive to distribution shifts and struggle to handle OOD objects absent from training data. Beyond traditional closed-set segmentation, recent advances in open-vocabulary 3D scene understanding (Peng et al., 2023), few-shot 3D segmentation (An et al., 2024b;a), and generalized segmentation with vision–language models (An et al., 2025) aim to improve flexibility and reduce reliance on large labeled datasets. However, these approaches are typically developed and evaluated on single datasets, and their generalization across diverse domains or unseen categories remains limited. This further motivates the need for a framework specifically designed to enhance both robustness and generalization in open-world 3D environments.

## A.2 MORE DETAILS ABOUT METHODOLOGY

### A.2.1 TEMPORAL-BASED FEATURE REGULARIZATION

We employ the widely used Temporal-based Feature Regularization module (Saltori et al., 2022; Chen & He, 2021) to mitigate negative transfer within the model. This process begins with calculating the temporal correspondences between feature points in the frames $X_T^{t-w}$ and $X_T^t$ by applying a rigid transformation $T^{t-w \to t}$ derived from the odometry data. Specifically, the temporal corresponding points $\Theta^{t-w \to t}$ are defined as:

$$\Theta^{t,t-w} = \left\{ \{x^t \in X_{\mathcal{T}}^t, x^{t-w} \in X_{\mathcal{T}}^{t-w}\} : x^t = \text{NN}\left(T^{t-w \to t}x^{t-w}, X_{\mathcal{T}}^t\right), \left\|x^t - x^{t-w}\right\|_2 < \tau' \right\} \tag{7}$$

Here, $x^t$ represents the nearest neighbor (NN$(,)$) point of $X_{\mathcal{T}}^t$ to the $T^{t-w \to t}x^{t-w}$, and $\tau'$ is a distance threshold that filters out pairs with distances above this value. Following SimSiam (Chen & He, 2021), we introduce an encoder network $h(.)$ and a predictor head $f(.)$ to the target model $F_T$. These components minimize the negative cosine similarity between the semantic representations of temporally corresponding points. The negative cosine similarity can be formulated as:

$$\mathcal{D}_{t \to t-w}\left(q^t, z^{t-w}\right) = -\frac{q^t}{\|q^t\|_2} \cdot \frac{z^{t-w}}{\|z^{t-w}\|_2} \tag{8}$$

where $z^t = h(x^t)$ is the encoder features, and $q^t = f(h(x^t))$ represents the predictor features. To enforce temporal consistency, we define the final temporal consistency loss as:

$$\mathcal{L}_{reg} = \frac{1}{2}\mathcal{D}_{t \to t-w}\left(q^t, z^{t-w}\right) + \frac{1}{2}\mathcal{D}_{t-w \to t}\left(q^{t-w}, z^t\right) \tag{9}$$

where `stop-grad` operator is applied to $z^t$ and $z^{t-w}$.

### A.2.2 FINAL ID PSEUDO-LABEL GENERATION

Given the ID pseudo-label $\hat{Y}_{ID}$ generated from the Temporal Pseudo-label Branch with the OOD pseudo-label $\hat{Y}_{OOD}$ derived from the Superpoint Representation Branch, we can refine $\hat{Y}_{ID}$ using information from $\hat{Y}_{OOD}$ and superpoints. We explore two optimization strategies: 1) **Direct Filtering:** We remove overlapping regions of $\hat{Y}_{OOD}$ from $\hat{Y}_{ID}$, resulting in a filtered ID pseudo-label, denoted as $\hat{Y}_{ID}/\hat{Y}_{OOD}$. 2) **Superpoint-Based Refinement:** Based on the filtered pseudo-label $\hat{Y}_{ID}/\hat{Y}_{OOD}$, we incorporate superpoints to further refine the pseudo-labels ($+Super$). Specifically, we assess each superpoint's ID pseudo-label composition. If a superpoint exclusively contains a single class of ID pseudo-labels, we assign all points within that superpoint to this class. If a superpoint contains multiple classes, it is excluded from further consideration.

As shown in Tab. 6, both strategies enhance open-set performance relative to the initial ID pseudo-label. Although Strategy Two yields a slight performance improvement over Strategy One, it risks introducing a substantial number of incorrect pseudo-labels, compromising its stability. Therefore, we select the simpler and more stable Strategy One.

### A.3 MORE EXPERIMENT DETAILS

#### A.3.1 SET UP

**Source and Target Datasets:** Following previous studies, we evaluate our method on the widely used Lidar datasets: **Synth4D** (Saltori et al., 2022), **SynLiDAR** (Xiao et al., 2021), **SemanticKITTI** (Behley et al., 2019) and **nuScenes** (Caesar et al., 2020). **Synth4D** (Saltori et al., 2022) is synthesized by the CARLA simulator (Dosovitskiy et al., 2017) and consists of 7 semantic classes, available in two configurations: 64-beams and 32-beams to simulate either SemantiKITTI or nuScenes. **SynLiDAR** (Xiao et al., 2021) is synthesized by the Unreal Engine 4 platform and contains 19 semantic classes to simulate SemantiKITTI. **SemanticKITTI** (Behley et al., 2019) is a large-scale real-world dataset for LiDAR point-cloud semantic segmentation, which contains 19 semantic classes. **nuScenes** (Caesar et al., 2020) has 32 class labels and 40K LiDAR frames annotated with per-point semantic labels from 1K sequences. Following the official sequence split, we use scene 08 in SemanticKITTI and 150 sequences in nuScenes for validation.

**Baseline Methods:** Due to the limitations of image-based TTA methods when applied to point clouds (Zou et al., 2024), we mainly compare our method with specific TTA-3DSeg baselines, such as **GIPSO** (Saltori et al., 2022) and **HGL** (Zou et al., 2024). Considering that the original TTA-3DSeg baselines are inadequate for addressing open-world scenarios, we incorporate several representative image-based OSTTA methods as an additional complement. **UniEnt** (Gao et al., 2024) adopts sample-based GMM to distinguish ID and OOD samples, followed by entropy minimization on the pseudo-ID data and entropy maximization on the pseudo-OOD data. **OSTTA** (Lee et al., 2023) uses the wisdom of crowds to filter out the samples with lower confidence values in the adapted model compared to the original model. In addition, we develop a straightforward method based on hyperparameter thresholds searching to distinguish between ID and OOD, termed Searching Thresholds (**SeaT**). This method targets high-uncertainty points in the pseudo-label generation process and classifies them as OOD samples.

**Implementation Details:** For the source model training, we use the Adam optimizer with an initial learning rate of 0.01, applying exponential decay, a batch size of 24, and a weight decay of $10^{-5}$. Training is conducted uniformly over 100 epochs across all source domain data, with the final model weights from the last epoch used as the definitive model. To avoid confusion between ID and OOD points, we employed the Dice loss instead of the soft Dice loss in GIPSO (Saltori et al., 2022). For online adaptation, we also apply the Adam optimizer with weight decay $10^{-5}$ and set the learning rate to $10^{-3}$ for all datasets. We avoid using learning rate schedulers to eliminate the need for prior knowledge about the length of the data stream. All experiments are conducted on L40 GPU with PyTorch-1.9.

**More Detail About RANSAC:** To ensure full reproducibility of preprocessing, each LiDAR scan is divided into 10×10 sub-regions to reduce the effect of uneven ground surfaces and allow RANSAC to operate on locally consistent regions, improving plane estimation accuracy. Candidate ground points are defined as those with height below -1 m relative to the ego-vehicle coordinate system, which works effectively across datasets due to the similar mounting height of vehicle LiDAR sensors. For RANSAC, the distance threshold is set to 0.1, the minimum number of points for model fitting is 3, and the number of iterations is 1000.

#### A.3.2 INTEGRATION INTO EXISTING METHODS

As elaborated in the main text of our study, the GOOD framework introduces an innovative approach to OSTTA-3DSeg through superpoint representation and temporal pseudo-label. Our approach can serve as a valuable complement to existing methods. To validate this merit, we have integrated GOOD with representative methods, specifically GIPSO (Saltori et al., 2022) and HGL (Zou et al., 2024). In light of the unsupervised nature of the OSTTA-3DSeg task, this integration is realized by aligning the optimization objectives of GOOD with those of the baseline methods. Specially, the integrated optimization objective is presented as follows:

$$\mathcal{L}_{overall} = \gamma \mathcal{L}_{GOOD} + (1 - \gamma)\mathcal{L}_{baseline} \tag{10}$$

where $\gamma$ is a trade-off hyper-parameter, generally set to 0.5. We can also combine the ID pseudo-labels from GOOD and baselines to obtain final pseudo-labels, which could yield similar results.

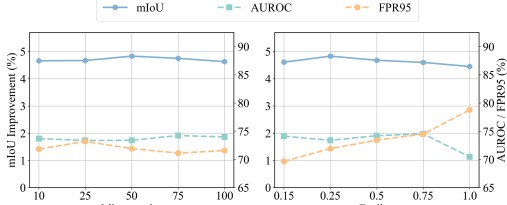 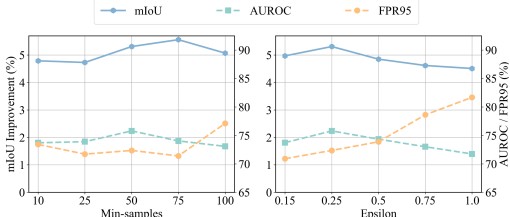

Figure 6: Hyper-parameter sensitivity analysis of DBSCAN on Synth4D to SemanticKITTI.

Figure 7: Hyper-parameter sensitivity analysis of DBSCAN on nuScenes to SemanticKITTI.

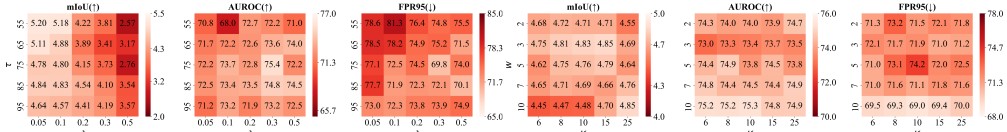

Figure 8: Highly coupled hyperparameters sensitivity analysis on Synth4D to SemanticKITTI.

### A.3.3 EXPERIMENTS ON INDOOR DATASETS

Our method is also applicable to indoor datasets. We conducted additional experiments on Scan-Net (Dai et al., 2017) using MinkUNet34C as the backbone. We use the standard ScanNet20 (Dai et al., 2017) classes as ID categories and treat all other classes in ScanNet200 (Rozenberszki et al., 2022) as OOD. The source model achieve mIoU 65.01%, AUROC 66.37%, and FPR95 88.93%. After introducing GOOD, the performance is mIoU 61.58%, AUROC 70.23%, and FPR95 85.93%. It can be observed that while the ID performance decreased, the OOD performance improved. The decline in ID performance is primarily due to due to the additional OOD loss, a phenomenon similarly reported in other 3D OS semantic segmentation studies (Cen et al., 2022; Li & Dong, 2023; Xu et al., 2023).

### A.3.4 RUNTIME ANALYSIS

We provide a detailed analysis of the runtime for Superpoint Clustering. In Superpoint Clustering, RANSAC takes approximately 0.55s on the CPU, while DBSACN clustering takes around 0.15s on the GPU and 0.25s on the CPU. Our method is slightly less efficient than HGL (0.65s) in speed but outperforms GIPSO (4.03s). For practical deployment, we offer two solutions to reduce runtime:

- Deploy Superpoint Clustering only on the CPU. CPU-based clustering takes about 0.8s, while TTA takes about 0.7s. In practice, clustering for the next frame can run concurrently with TTA.
- Replace RANSAC with a height threshold (e.g., -1.4m (Geiger et al., 2012)), sacrificing a small amount of perception accuracy (mIoU(↑) from +4.83% to +4.65%, AUROC(↑) from 73.39% to 71.50%, FPR95(↓) from 71.91% to 77.39%) but achieving a runtime comparable to HGL(0.86s vs 0.65s).

### A.3.5 MORE ABLATION ANALYSIS

We conduct a more comprehensive analysis in GOOD. Unless otherwise specified, the ablation analysis is based on Synth4D to SemanticKITTI.

**Different Superpoint Clustering:** We also experimented with clustering algorithms other than DB-SCAN, including K-Means and the Sinkhorn-Knopp-based optimal transport algorithm, as shown in Tab. 7. Compared to DBSCAN, K-Means and Sinkhorn-Knopp require manual setting of the number of clusters, which limits their adaptability. Moreover, they cluster all points without filtering, resulting in the inclusion of a large number of outliers. Due to these two factors, their performance is inferior. It is also worth noting that unlike CPCR's online feature clustering (Feng et al., 2023), the OT offline coordinate clustering is significantly slower than the DBSCAN algorithm (50.30s vs 0.25s). While DBSCAN has a time complexity of $O(N^2)$, OT's worst-case complexity is approximately $O(TNMt)$, where T, N, M, and t denote the number of center iterations, points, clusters, and Sinkhorn iterations, respectively. Given $T = 100, N = 100000, M = 100$ and $t = 1000$, OT is up to 100 times slower than DBSCAN.

Table 6: Results on different ID pseudo-label strategy.

| Model | $\hat{Y}_{ID}$ | $\hat{Y}_{ID}/\hat{Y}_{OOD}$ | $(\hat{Y}_{ID}/\hat{Y}_{OOD}) + Super$ |
|---|---|---|---|
| mIoU(%) | +4.95 ± 0.07 | +4.83 ± 0.06 | +4.87 ± 0.19 |
| AUROC(%) | 72.49 ± 0.36 | 73.39 ± 0.37 | 73.40 ± 0.43 |
| FPR95(%) | 76.29 ± 2.25 | 71.91 ± 1.09 | 72.50 ± 1.57 |

Table 7: Results on different superpoint clustering.

| | mIoU(%) | AUROC(%) | FPR95(%) | Runtime(s) |
|---|---|---|---|---|
| Kmeans | +4.30 | 70.75 | 74.42 | 0.18 |
| Sinkhorn-Knopp | +4.26 | 69.25 | 74.60 | 50.30 |
| DBSCAN | +4.83 | 73.39 | 71.91 | 0.25 |

Table 8: Ablation Study. **TP**: Temporal Pseudo-label. **TR**: Temporal Regularization. **SC**: Superpoint Confidence. **SIDP**: Superpoint ID Prototypes. **EMA**: Exponential Moving Average. *: without Temporal Regularization.

| ID | TP | TR | SC | SIDP | EMA | mIoU(%) | AUROC(%) | FPR95(%) | Rumtime(s) |
|---|---|---|---|---|---|---|---|---|---|
| Source | - | - | - | - | - | 40.26 | 65.80 | 78.39 | - |
| GIPSO (Saltori et al., 2022) | - | ✓ | - | - | - | +2.08 | 64.47 | 80.40 | 4.03 |
| GIPSO* (Saltori et al., 2022) | - | - | - | - | - | -0.26 | 63.30 | 90.73 | 3.85 |
| HGL (Zou et al., 2024) | - | ✓ | - | - | - | +2.90 | 64.40 | 79.82 | 0.65 |
| HGL* (Zou et al., 2024) | - | - | - | - | - | +0.51 | 63.01 | 88.28 | 0.46 |
| I | ✓ | - | - | - | - | +2.64 | 65.36 | 87.02 | 0.52 |
| II | ✓ | ✓ | - | - | - | +3.36 | 67.09 | 80.47 | 0.70 |
| III | ✓ | ✓ | ✓ | - | - | +3.56 | 73.05 | 73.96 | 1.38 |
| IV | ✓ | ✓ | ✓ | ✓ | - | +3.69 | 75.72 | 71.54 | 1.42 |
| V | ✓ | - | ✓ | ✓ | ✓ | +3.07 | 74.28 | 72.91 | 1.30 |
| VI | ✓ | ✓ | - | - | ✓ | +4.61 | 66.66 | 79.72 | 0.70 |
| VII | ✓ | ✓ | ✓ | - | ✓ | +4.54 | 71.47 | 76.82 | 1.42 |
| VIII | ✓ | ✓ | ✓ | ✓ | ✓ | +4.83 | 73.39 | 71.91 | 1.45 |

**More Detail About DBSCAN:** DBSCAN is a popular density-based clustering algorithm that identifies clusters by grouping points within proximity and designating points in low-density areas as outliers. The algorithm relies on two key parameters: epsilon and min-samples. Epsilon defines the maximum distance between two points for them to be considered part of the same neighborhood, while min-samples is the minimum number of points required to form a cluster. In our experiments, we set epsilon as 0.25 and set min-samples as 50 for SemanticKITTI and 25 for nuScenes.

We first examine the effect of min-sample, with epsilon taking values from {10, 25, 50, 75, 100} and epsilon holds constant. The experimental results in the left side of Fig. 6 show that performance remains stable around the chosen parameter value of min-samples = 50. Next, we investigate the impact of varying epsilon on OOD superpoint detection, holding min-samples constant. We test epsilon values from {0.15, 0.25, 0.50, 0.75, 1.00}, as illustrated on the right side of Fig. 6. The results reveal that epsilon is more sensitive than min-samples. Notably, open-set performance decreases as epsilon increases, while closed-set performance initially improves and subsequently declines. This behavior occurs because a low epsilon value leads to more discrete and smaller superpoints, enhancing the distinction between ID and OOD suerpoints. However, this also makes it more challenging for ID pseudo-labels to benefit from the OOD pseudo-labels. In contrast, when epsilon is higher, the superpoints expand and may encompass multiple classes, which can increase misclassification rates. Therefore, a moderate epsilon value, such as 0.25, balances the trade-off between superpoint granularity and classification accuracy, yielding satisfactory performance in both open-set and closed-set scenarios. In addition to the sim-to-real experiments presented in Fig. 6, we further conduct real-to-real experiments as shown in Fig. 7. The results consistently demonstrate the effectiveness and stability of the proposed hyperparameters.

**More Hyper-parameter Sensitivity Analysis:** We perform detailed sensitivity analyses on two strongly coupled hyperparameter groups: (1) $K$ in K-NN and the $w$ in temporal aggregation for temporal information, and (2) confidence threshold $\tau$ and the loss weight $\lambda$ for OOD separation. Fig. 8 shows stable performance within selected ranges. We further conduct a sensitivity analysis on the EMA hyperparameter $\alpha$, as shown in Tab. 9. In experiments on the Synth4D-to-SemanticKITTI setting, GOOD shows low sensitivity to this hyperparameter. In highly dynamic scenarios with large domain shifts, a smaller $\alpha$ can help maintain stability.

**More Component Analysis:** As shown in Tab. 8, we conduct a more comprehensive analysis of the components in GOOD, where components Temporal Pseudo-label (TP), Superpoint Confidence (SC), and Superpoint ID Prototypes (SIDP) have been previously discussed in the main text. Based on the results of experiments IV and VIII, we observe that incorporating Exponential Moving Average (EMA) significantly enhances closed-set performance, but slightly reduces open-set performance. We hypothesize that this trade-off arises because EMA improves ID pseudo-label accuracy through consistency, which may inadvertently propagate the prior overconfidence of the source model regarding OOD points, thereby negatively impacting the OOD pseudo-label. Experiments I, II, V, and VIII demonstrate that Temporal Regularization (TR) effectively improves both the efficiency and stability of the model's closed-set performance, also reported in (Saltori et al., 2022; Zou

Table 9: Hyper-parameter sensitivity analysis of the EMA coefficient $\alpha$.

| $\alpha$ | 0.9999 | 0.999 | 0.99 | 0.9 | 0.7 |
|---|---|---|---|---|---|
| mIoU($\uparrow$) | +4.70 | +4.78 | +4.83 | +4.67 | +4.66 |
| AUROC($\uparrow$) | 72.42 | 73.26 | 73.39 | 73.97 | 74.49 |
| FPR95($\downarrow$) | 74.62 | 72.40 | 71.91 | 71.16 | 70.50 |

Table 10: Results w/ and w/o SIDP under varying OOD ratios.

| OOD ratios | mIoU($\uparrow$), AUROC($\uparrow$), FPR95($\downarrow$) | | |
|---|---|---|---|
| | Source | GOOD w/o SIDP | GOOD |
| $\sim 0.1\%$ | (26.21, 60.42, 94.74) | (+3.87, 56.26, 95.77) | (**+4.93, 62.65, 91.39**) |
| $\sim 6.4\%$ | (24.93, 66.23, 70.18) | (+3.44, 70.44, 69.25) | (**+3.60, 71.90, 65.75**) |
| $\sim 12.9\%$ | (46.48, 64.02, 90.47) | (+2.03, 74.40, 76.67) | (**+2.68, 74.64, 76.33**) |

Table 11: Results on different superpoint confidence.

| Confidence | MSP | Margin | Purity | Entropy | Purity+Entropy |
|---|---|---|---|---|---|
| mIoU(%) | +4.89 | +5.05 | +5.21 | +4.00 | +4.83 |
| AUROC(%) | 69.79 | 69.72 | 70.19 | 74.86 | 73.39 |
| FPR95(%) | 77.22 | 73.13 | 75.43 | 69.74 | 71.91 |

Table 12: mIoU (%) results on close-set.

| Model | Synth4D2KITTI | Synth4D2nuSc | Synlidar2KITTI |
|---|---|---|---|
| Source | 35.93 | 30.75 | 40.19 |
| HGL | +6.40 | +1.87 | +6.72 |
| Ours | +7.05 | +2.22 | +6.54 |

et al., 2024). Additionally, compared to previous pseudo-label generation methods (Saltori et al., 2022; Zou et al., 2024), our temporal pseudo-label approach proves to be more robust to regularization due to its incorporation of temporal consistency, which effectively mitigates negative transfer during optimization.

**Prototype Stability and Safeguards:** In our framework, prototypes are updated via EMA using pseudo-labeled superpoints, which raises the question of whether early-stage noise may cause accumulated drift. To investigate this, we compared several safeguard mechanisms, including weighted prototypes (WP) Manna et al. (2024), threshold-based prototypes (TP) Su et al. (2024), and multi-center prototypes (MP) Qu et al. (2022). The results in Tab. 13 show that WP achieves performance comparable to the original approach, TP performs slightly worse, likely due to strict filtering of valuable samples, and MP shows mixed effects, improving some metrics while hurting others. The limited benefit of TP and MP can be attributed to the prototype construction process: superpoints are aggregated, and a GMM selects only high-confidence samples, mitigating the influence of noisy inputs at an early stage. Additional safeguards introduce extra hyperparameters without clear gains, so the original prototype formulation is retained for both simplicity and effectiveness. To further assess temporal stability, classification accuracy for known and unknown categories was monitored before and after prototype updates, aggregated every 500 training steps, yielding values [1.61%, 1.85%, 2.57%, 2.37%, 3.83%, 1.41%, 2.52%, 4.86%]. These results show an overall upward trend, indicating that prototype updates stabilize over time and reduce category assignment errors rather than amplifying early-stage noise. Together, these analyses confirm that the baseline prototype mechanism is robust and temporally stable.

**Loss Function Analysis:** We compared multiple loss functions for ID pseudo-label learning, including Dice loss, Soft Dice loss, SCE loss, and CE loss. The results are summarized in Tab. 14. Soft Dice loss leads to severe degradation in open-set detection capability, reflected in lower AUROC and higher FPR95, while SCE and CE losses achieve strong open-set results but insufficient closed-set segmentation performance, indicating that these losses overemphasize separability at the cost of reconstruction quality. Dice loss provides the best balance across both closed-set and open-set tasks. Its effectiveness for small objects and boundary regions, along with inherent normalization that mitigates class imbalance, makes it particularly suitable for superpoint aggregation and prototype construction, supporting both stable pseudo-label refinement and consistent adaptation across domains.

**Alternative Strategies to GMM:** We also investigated threshold-based strategies that separate known and unknown categories using manually selected thresholds on purity and entropy. Two variants were evaluated: threshold-open, which favors open-set performance, and threshold-closed, which favors closed-set performance. Although these approaches achieve reasonable results, they require additional hyperparameter tuning, and optimal thresholds vary across datasets, making stable performance difficult without dataset-specific adjustment. In contrast, the GMM-based method adapts automatically to the score distribution of each domain and does not require manual thresholds. Empirical results, summarized in Tab. 15, show that while threshold-based strategies can achieve competitive metrics, the GMM-based method provides a more robust and dataset-independent solution, motivating its use as the default.

**Different Backbone Architectures:** To assess the architectural generality of our framework, we evaluate its performance on a diverse set of backbone models, including MinkowskiUNet-14 Choy et al. (2019), MinkowskiUNet-18 Choy et al. (2019), MinkowskiUNet-34 Choy et al. (2019), Point-Net++ Qi et al. (2017b), and PTv2 Wu et al. (2022), covering voxel-based, point-based, and hybrid designs. As shown in Tab. 16, the method consistently improves both closed-set mIoU and open-

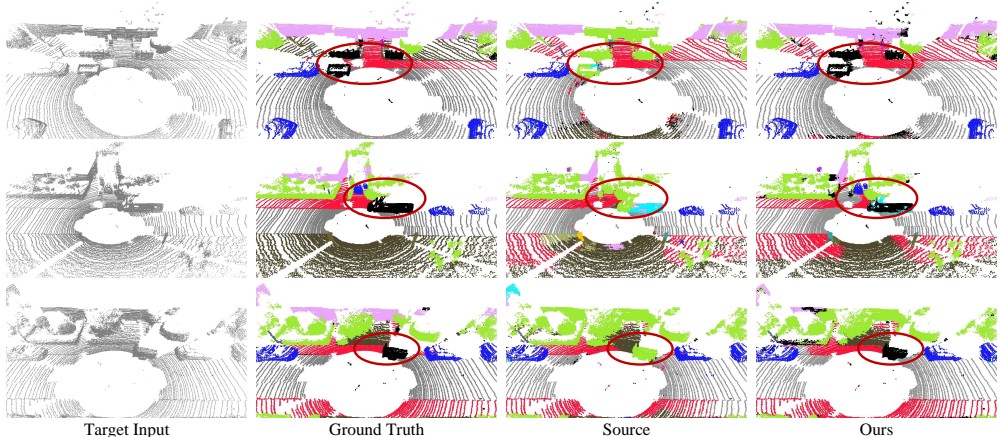

Figure 9: Qualitative results in SynLiDAR to SemanticKITTI. Black pixels indicate the open-set points, and we consistently select points with Maximum Softmax Probability (MSP) greater than 0.6 as OOD points.

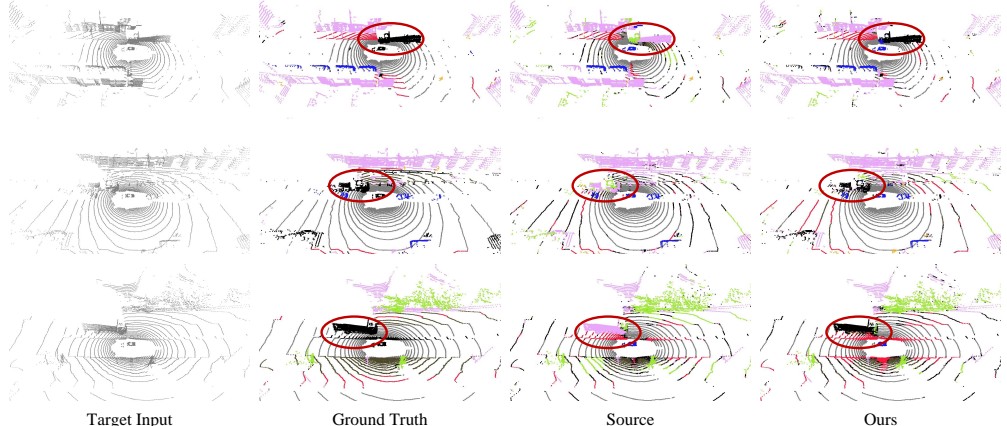

Figure 10: Qualitative results in Synth4D to nuScenes.

set AUROC/FPR95 for all backbones, demonstrating that the proposed approach is architecture-agnostic and broadly compatible with different 3D semantic segmentation models.

**Different Superpoint Confidence:** We experiment with different confidence metrics in the super-point representation, including Maximum Softmax Probability (MSP) (Hendrycks & Gimpel, 2016), Margin (Roth & Small, 2006), Purity (Zou et al., 2024), and Entropy (Shannon, 1948). As shown in Tab. 11, Purity outperforms MSP and Margin, though all three demonstrate suboptimal open-set performance. While Entropy shows limited efficacy in the closed-set setting, it achieves outstanding results in open-set scenarios. Based on these observations, we select Entropy to complement Purity, yielding robust performance in both open and closed sets. With GT labels, we also quantitatively report the average confidence scores of Purity+Entropy for ID and OOD superpoints as 0.57 and 0.81, respectively, validating the effectiveness of our intuition in Sec. 3.2.

**Performance under Close-set:** Although our method is designed for open-world scenarios, previous experiments have indicated that the ID pseudo-labels generated by the Temporal Pseudo-labels Branch exhibit high quality, while the OOD pseudo-labels produced by the Superpoint Representation Branch can be regarded as an additional filter for the pseudo-labels. Consequently, we attempt to evaluate our method in a purely close-set. To accommodate the closed-set configuration, we removed $L_{OOD}$ and substituted $L_{dice}$ with $L_{soft\_dice}$ to ensure a fair comparison. As shown in Tab. 12, our method achieves performance comparable to the state-of-the-art approach in Synth4D2KITTI and attains state-of-the-art results in Synth4D2nuSc and Synlidar2KITTI. These results demonstrate that GOOD exhibits strong generalizability and is effective in closed sets.

**More Sparse OOD Scenario:** We also conducted experiments on the SynLiDAR to SemanticKITTI dataset, controlling the data ratio at 0.1%, with *person* designated as an unknown class, representing a more sparse OOD scenario. We observe AUROC improvement from 60.42% to 62.65% and

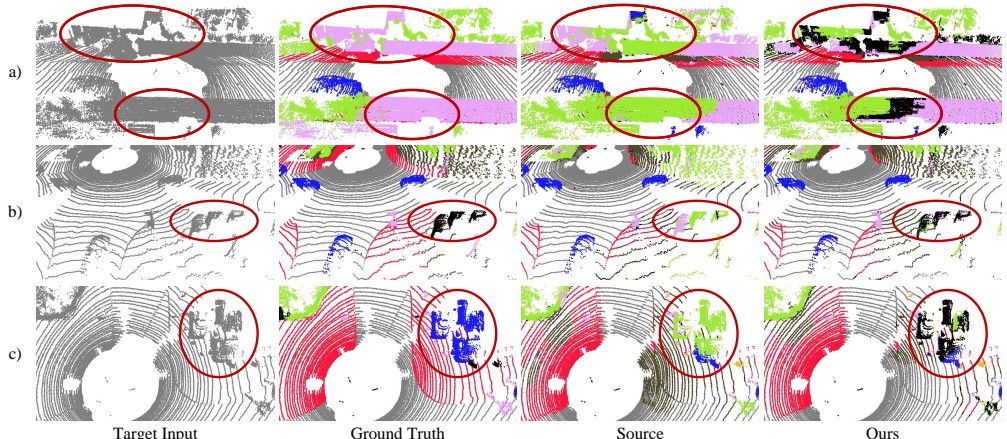

Target Input        Ground Truth        Source        Ours

Figure 11: Failure cases in Synth4D to SemanticKITTI. Black pixels indicate the open-set points, and we consistently select points with Maximum Softmax Probability (MSP) greater than 0.6 as OOD points.

Table 13: Results on different prototype safeguard mechanisms

|  | Original | WP | TP | MP |
|---|---|---|---|---|
| mIoU (%) | +4.83 | +4.81 | +4.57 | +4.94 |
| AUROC (%) | 73.39 | 74.01 | 72.87 | 72.72 |
| FPR95 (%) | 71.91 | 71.38 | 75.76 | 76.40 |

Table 14: Results on different ID pseudo-label learning losses.

|  | Source | Dice(GOOD) | SoftDice | SCE | CE |
|---|---|---|---|---|---|
| mIoU (%) | 40.26 | +4.83 | +4.81 | +3.33 | +3.53 |
| AUROC (%) | 65.80 | 73.39 | 68.62 | 76.70 | 75.70 |
| FPR95 (%) | 78.39 | 71.91 | 84.87 | 63.26 | 64.07 |

FRP95 improvement from 94.74% to 91.39%, where SIDP plays a pivotal role. Without SIDP, both the AUROC and FPR95 exhibit a decline in performance (from 60.42% to 60.26% and 94.74% to 95.77%). Furthermore, we show that SC may fail under extreme imbalance where OOD points are extremely sparse or entirely absent, whereas our SIDP strategy effectively alleviates this issue. The Tab. 10 presents the results with and without the SIDP strategy across a range of OOD ratios in the SynLiDAR-to-SemanticKITTI setting, demonstrating the effectiveness of SIDP in mitigating GMM's over-partitioning behavior.

### A.3.6 DISCUSSION

**Combine with Open-set Segmentation:** In addition to combining the existing TTA-3DSeg and OSTTA-2D, we can also directly use 3D OS semantic segmentation (3D-OSSS) as a baseline. Considering code reproduction and feasibility, we use LiON (Xu et al., 2023) as the baseline. Compared to the normal source model, which yields an mIoU of 26.95%, AUROC of 54.90%, and FPR95 of 86.67% on SynLiDAR to SemanticKITTI, LiON produces an mIoU of 25.44%, AUROC of 65.47%, and FPR95 of 75.19%. Although 3D-OSSS performs better, it is important to note that 3D-OSSS (Cen et al., 2022; Li & Dong, 2023; Xu et al., 2023) alters the source model's network structure, training strategy, and even introduces additional OOD datasets, which contradicts the goal of TTA methods to avoid retraining the original model. Therefore, we still choose the combination of TTA-3DSeg and OSTTA-2D as the baseline model to enhance generalizability. Additionally, 3D-OSSS and our method are not in competition but are complementary. The GOOD approach can also be applied to the source model obtained through 3D-OSSS. After applying GOOD, the LiON results improved to an mIoU of 30.53%, AUROC of 77.66%, and FPR95 of 66.85%.

**Ego-Pose Assumption and Robustness Analysis:** Following standard practice in prior works (Saltori et al., 2022; Zou et al., 2024), our framework assumes access to reliable ego poses for temporal alignment; however, we acknowledge that pose quality may vary in real-world applications, and therefore we provide extensive analyses to evaluate robustness under imperfect conditions. To verify whether GOOD critically relies on perfect poses, we replace the provided ego poses with estimates obtained through GICP Segal et al. (2009), a fast and accurate registration-based method for dense geometric alignment. The results in Tab. 17 show minimal performance differences compared with using perfect poses, demonstrating that temporal aggregation remains effective with registration-derived alignment. To further quantify sensitivity, we inject controlled translation and rotation perturbations into the poses. As shown in Tab. 19, we find that while trans-

Table 15: Comparison of GMM and threshold-based strategies

|  | Source | GMM | Threshold-open | Threshold-closed |
|---|---|---|---|---|
| mIoU (%) | 40.26 | +4.83 | +4.43 | +4.90 |
| AUROC (%) | 65.80 | 73.39 | 74.46 | 73.77 |
| FPR95 (%) | 78.39 | 71.91 | 70.55 | 73.01 |

Table 16: Results on different backbone architectures.

|  | PointNet++ | | Mk-14 | | Mk-18 | | Mk-34 | | PTv2 | |
|---|---|---|---|---|---|---|---|---|---|---|
|  | Source | Ours | Source | Ours | Source | Ours | Source | Ours | Source | Ours |
| mIoU | 26.51 | +6.80 | 35.76 | +7.00 | 40.26 | +4.83 | 42.79 | +4.80 | 44.45 | +4.27 |
| AUROC | 53.05 | 67.35 | 64.90 | 75.03 | 65.80 | 73.39 | 63.16 | 72.15 | 63.71 | 78.38 |
| FPR95 | 95.47 | 81.16 | 85.30 | 76.85 | 78.39 | 71.91 | 87.70 | 79.66 | 85.21 | 70.03 |

Table 17: Performance comparison under different pose settings.

|  | Source | Perfect Pose | GICP Pose |
|---|---|---|---|
| mIoU (%) | 40.26 | +4.83 | +4.80 |
| AUROC (%) | 65.80 | 73.39 | 74.29 |
| FPR95 (%) | 78.39 | 71.91 | 71.23 |

Table 18: Performance comparison of GOOD-single and different models.

|  | Source | GOOD | GOOD-single | HGL | GIPSO |
|---|---|---|---|---|---|
| mIoU (%) | 40.26 | +4.83 | +2.37 | +2.90 | +2.08 |
| AUROC (%) | 65.80 | 73.39 | 73.23 | 64.40 | 64.47 |
| FPR95 (%) | 78.39 | 71.91 | 75.04 | 79.82 | 80.40 |

Table 19: Results under different transformations and rotations perturbations. Translation noise is denoted as $T_{err}$, where the noise is uniformly sampled within $[-X, X]$ meters, and rotation noise is denoted as $R_{err}$, sampled within $[-Y, Y]$ degrees.

|  | Source | Perfect Pose | $T_{0.5}$-$R_0$ | $T_{1.0}$-$R_0$ | $T_{2.0}$-$R_0$ | $T_{0.0}$-$R_5$ | $T_{0.0}$-$R_{10}$ | $T_{0.0}$-$R_{20}$ | $T_{0.5}$-$R_5$ | $T_{1.0}$-$R_{10}$ | $T_{2.0}$-$R_{20}$ |
|---|---|---|---|---|---|---|---|---|---|---|---|
| mIoU (%) | 40.26 | **+4.83** | +4.54 | +4.29 | +2.86 | +1.75 | +0.91 | +0.51 | +0.98 | +0.55 | -0.08 |
| AUROC (%) | 65.80 | **73.39** | 73.28 | 73.94 | 74.53 | 74.43 | 72.27 | 72.76 | 73.31 | 72.41 | 70.35 |
| FPR95 (%) | 78.39 | **71.91** | 73.55 | 74.92 | 72.39 | 68.95 | 77.52 | 74.21 | 75.11 | 78.05 | 77.84 |

lation noise has limited influence, rotation noise leads to more noticeable degradation, and very large combined perturbations (e.g., 2 m and 20°) cause temporal correspondences to collapse, eliminating the gains from temporal consistency, which is expected given the correspondence-driven nature of TPB. To address scenarios where pose estimates are unreliable or entirely unavailable, we additionally evaluate a single-frame variant of GOOD that removes temporal consistency and relies solely on intra-frame superpoint aggregation and pseudo-labeling. As shown in Tab. 18, although its performance is lower than the full GOOD pipeline, it still surpasses GIPSO in closed-set accuracy and remains competitive or superior in open-set metrics, indicating that meaningful adaptation is achievable even without temporal alignment. Collectively, these results confirm that GOOD is not strongly dependent on perfect ego poses: temporal cues improve performance when available, registration-based alignment provides a practical fallback, and the single-frame variant remains a robust solution in the absence of reliable pose information.

**Removal of Ground:** The reason for removing ground points is that, generally, flat surfaces like roads or sidewalks are not considered OOD, as nearly all virtual source data contains these categories. Our primary focus is on rare OOD objects above the ground, which could pose significant safety risks (e.g., skateboards, wheelchairs).

**Failure Cases:** We present typical failure cases in Fig. 11. (1) Due to errors in the source model's predictions, our model's correction attempts result in lower confidence scores. Additionally, some edge points also exhibit low confidence. As shown in the red circle in Fig. 11 a), the source model incorrectly predicts *manmade* as *vegetation*, and GOOD misclassifies it as OOD while attempting to correct this error. (2) Due to point-wise prediction, some points within the same OOD object may be classified as ID, causing confusion, as shown in Fig. 11 b). (3) Some ID objects exhibit significant variation, leading to misclassification as OOD. As shown in Fig. 11 c), a *vehicle* is misclassified due to its similarity to the unknown *truck*. In future work, we can explore methods such as edge loss and introducing a memory bank to mitigate these cases.

## A.4 VISUALIZATION RESULTS

In Fig. 9 and 10, we report additional adaptation visualization results of the GOOD, source, and ground truth in SynthLiDAR to SemanticKITTI and Synth4D to nuScenes. We observe that our method significantly improves both close-set and open-set performances.

## A.5 CLASS MAPPING

Due to the different annotations across datasets, we follow GIPSO (Saltori et al., 2022) to remap semantic categories of SemanticKITTI (Behley et al., 2019) and nuScenes (Caesar et al., 2020) into seven standardized categories in Synth4D (Saltori et al., 2022) to ensure consistent category defini-

Table 20: Class mapping from SemanticKITTI to Synth4D.

| SemanticKITTI-ID | SemanticKITTI-Name | Synth4D-Name | Synth4D-ID |
|---|---|---|---|
| 0 | unlabelled | unlabelled | 0 |
| 1 | car | vehicle | 1 |
| 2 | bicycle | unlabelled | 0 |
| 3 | motorcycle | unlabelled | 0 |
| 4 | truck | unlabelled | 0 |
| 5 | other-vehicle | unlabelled | 0 |
| 6 | person | pedestrian | 2 |
| 7 | bicyclist | unlabelled | 0 |
| 8 | motorcyclist | unlabelled | 0 |
| 9 | road | road | 3 |
| 10 | parking | road | 3 |
| 11 | sidewalk | sidewalk | 4 |
| 12 | other-ground | unlabelled | 0 |
| 13 | building | manmade | 6 |
| 14 | fence | manmade | 6 |
| 15 | vegetation | vegetation | 7 |
| 16 | trunk | vegetation | 7 |
| 17 | terrain | terrain | 5 |
| 18 | pole | manmade | 6 |
| 19 | traffic-sign | manmade | 6 |

Table 21: Class mapping from nuScenes to Synth4D.

| nuScenes-ID | nuScenes-Name | Synth4D-Name | Synth4D-ID |
|---|---|---|---|
| 0 | unlabelled | unlabelled | 0 |
| 1 | barrier | unlabelled | 0 |
| 2 | bicycle | unlabelled | 0 |
| 3 | bus | unlabelled | 0 |
| 4 | car | vehicle | 1 |
| 5 | construction-vehicle | unlabelled | 0 |
| 6 | motorcycle | unlabelled | 0 |
| 7 | pedestrian | pedestrian | 2 |
| 8 | traffic-cone | unlabelled | 0 |
| 9 | trailer | unlabelled | 0 |
| 10 | truck | unlabelled | 0 |
| 11 | driveable-surface | road | 3 |
| 12 | other-flat | unlabelled | 0 |
| 13 | sidewalk | sidewalk | 4 |
| 14 | terrain | terrain | 5 |
| 15 | manmade | manmade | 6 |
| 16 | vegetation | vegetation | 7 |

tions. Tab. 20 and 21 show the class mapping from SemanticKITTI to Synth4D and nuScenes to Synth4D, respectively. Categories that do not intersect with other datasets are considered unlabeled.

## A.6 LARGE LANGUAGE MODELS ACKNOWLEDGMENT

We employed Large Language Models (LLMs) during the manuscript writing process to assist with and refine the writing. Specifically, LLMs were used to optimize the fluency and phrasing of the text, enhancing both linguistic accuracy and readability. Additionally, the models were leveraged to improve the structure of certain sections, contributing to a clearer overall logical flow of the paper.

