# OpenReview forum: "GOOD: Geometry-guided Out-of-Distribution Modeling for Open-set Test-time Adaptation in Point Cloud Semantic Segmentation"
_ICLR.cc/2026/Conference — ICLR 2026 Poster_

### Official Review · Reviewer_XJDX · 2025-10-19

**Soundness:** 3
**Presentation:** 3
**Contribution:** 3
**Rating:** 6
**Confidence:** 4

**Summary:**

This paper proposes GOOD, a framework for open-set test-time adaptation (OSTTA) in 3D point cloud semantic segmentation. GOOD addresses the imbalance between in-distribution (ID) and out-of-distribution (OOD) points by introducing two key branches: the Superpoint Representation Branch (SRB) and the Temporal Pseudo-label Branch (TPB). Experiments demonstrate that GOOD achieves superior results compared to existing methods.

**Strengths:**

1. The paper is clearly written, and the figures are effective in illustrating the technical flow.
2. The design of SRB and TPB is technically sound: SRB clusters point clouds into superpoints and defines confidence metrics to better distinguish ID and OOD regions, while TPB enforces temporal consistency across sequential frames to produce more stable pseudo-labels.
3. The experimental evaluation is extensive, providing strong evidence of the method’s effectiveness.

**Weaknesses:**

1. For 3D open-set segmentation, there are models such as OpenScene that leverage open-vocabulary queries to achieve open-set segmentation. A comparison with such models would be valuable. In particular, how would be the performance difference, and what are the differences and benefits of adopting an open-set test-time adaptation approach compared to the open-vocabulary segmentation paradigm?
2. Missing related work. Several recent works on 3D segmentation are relevant and should be cited:
   - OpenScene: 3D Scene Understanding with Open Vocabularies (CVPR 2023)
   - Rethinking few-shot 3d point cloud semantic segmentation (CVPR 2024)
   - Multimodality Helps Few-shot 3D Point Cloud Semantic Segmentation (ICLR 2025)
   - Generalized Few-shot 3D Point Cloud Segmentation with Vision-Language Model (CVPR 2025)

**Questions:**

Please refer to the weakness part and address the concerns there.

---

> ### Author Response · Authors · 2025-11-20
> **Responces to Reviewer XJDX**
>
> We sincerely thank reviewer XJDX for your thoughtful feedback on our work and hope all concerns have been addressed. We will update our manuscript in a few days.
>
> **W1: Comparison with open-vocabulary 3D segmentation methods.**
>
> **A1:** Thank you for raising this important point. We conducted additional analysis using OpenScene [1], a representative open-vocabulary 3D segmentation model, to evaluate its applicability in our OSTTA setting. We directly transferred the OpenScene model pretrained on nuScenes to the SemanticKITTI dataset. Neverthless, the results were not satisfactory: the overall mIoU dropped to 8%, with only the “road” category achieving an mIoU of 25%, while the performance across all other categories remained uniformly low.
>
> We attribute this limitation to several factors: (1) OpenScene distills image-based open-vocabulary segmentation models, which rely heavily on visual–text alignment learned from 2D corpora. This modality gap makes generalization to 3D-only domains challenging; (2) Its training and evaluation are restricted to a single dataset, resulting in limited cross-domain generalization capability. (3) On the nuScenes dataset itself, its open-vocabulary performance is significantly lower than fully supervised baselines, indicating that the distilled representations are not strong enough for robust 3D segmentation. (4) OpenScene requires heavy text-embedding fine-tuning (42% mIoU with tuning versus 30% without tuning).
>
> We also examined 3DOSSS [2], another open-set or open-vocabulary alternative. However, 3DOSSS is similarly designed and evaluated on a single dataset, and it does not explicitly consider test-time adaptation or domain shifts. We discuss its properties further in Supplementary Material A.3.6 (“Combine with Open-set Segmentation”). Importantly, open-vocabulary 3D segmentation and our method are not mutually exclusive.
> Our framework can serve as a complementary module that enhances the cross-domain generalization of open-vocabulary methods by providing adaptation at test time.
>
> > [1] Peng, Songyou, et al. "Openscene: 3d scene understanding with open vocabularies." CVPR 2023.
> >
> > [2] Xu, Shaocong, et al. "LiON: Learning Point-wise Abstaining Penalty for LiDAR Outlier DetectioN Using Diverse Synthetic Data." AAAI 2025.
>
>
> **W2: Missing related work.**
>
> **A2:** Thank you for pointing out these omissions. We will add the suggested references to the revised manuscript.

---

### Official Review · Reviewer_sTXL · 2025-10-21

**Soundness:** 3
**Presentation:** 3
**Contribution:** 3
**Rating:** 6
**Confidence:** 4

**Summary:**

This paper proposes to more effectively distinguish ID and OOD samples through indicators such as superpoint confidence and purity, as well as ID superpoint prototypes. Furthermore, it designs a temporal pseudo-label method to generate more accurate pseudo-labels. Experimental results show that it achieves optimal performance under the open-set TTA setting, which demonstrates the effectiveness of its method.

**Strengths:**

1.This paper achieves a better distinction between ID and OOD samples through superpoint clustering and the utilization of superpoints, which is well demonstrated by visualization. Therefore, it exhibits strong innovation and rationality.

2.
The paper's method is simple and efficient, and can be applied to many point cloud segmentation methods.



3.The paper's experiments and visualizations are comprehensive, and the results all demonstrate the effectiveness of its method.

**Weaknesses:**

1.The paper reports the running time but fails to report the computational overhead such as GFLOPs.

2.The definition of symbols in the formulas is rather confusing. For example, the definition of the subscript of y$_{j,c}$  in Eq. 1 is not clearly explained.

**Questions:**

1.Is it possible to add more experiments with different backbones to demonstrate the generality of the method?

2.In Table 4, the increase in mIOU metric when adding more components is not significant. Could the reasons for this be analyzed?

3.The impact of noisy samples be considered when constructing prototypes? This is quite common in TTA methods for 2D images.

4.In Table 4, does " superpoint representation (SR)" refer to the use of superpoint purity and confidence in GMM? It should be explained in more detail.

---

> ### Author Response · Authors · 2025-11-20
> **Responces to Reviewer sTXL [Part-1]**
>
> We sincerely thank reviewer sTXL for your thoughtful feedback on our work and hope all concerns have been addressed. Unless otherwise noted, all experiments are conducted on Synth4D and evaluated on SemanticKITTI. We will update our manuscript in a few days.
>
> **W1: Computational cost.**
>
> **A1:** Thans for your comment. Most existing works [1, 2, 3] report computational overhead in GFLOPs as a measure of computational cost. However, GFLOPs only capture the forward computation of the network and do not reflect the full workload of test-time adaptation. In the TTA setting, the pipeline includes both inference and online optimization, and these additional optimization steps cannot be meaningfully represented using GFLOPs, since they involve operations that fall outside the standard network forward path.
>
> Furthermore, our method includes non-learning procedures such as DBSCAN and geometry-based clustering, for which GFLOPs are not an appropriate or informative metric because their complexity depends on point density and neighborhood queries rather than neural network computation. For this reason, we report wall-clock inference time under the same hardware configuration, which provides a more practical and comparable indication of computational cost across TTA methods. We will clarify this rationale in the revised manuscript.
>
> > [1] Fei, Zhengcong, et al. "Scalable diffusion models with state space backbone." arXiv 2024.
> >
> > [2] Zhang, David Junhao, et al. "Morphmlp: An efficient mlp-like backbone for spatial-temporal representation learning.". ECCV 2022
> >
> > [3] Peebles, William, et al. "Scalable diffusion models with transformers." ICCV 2023
>
>
> **W2: Symbol definitions in formulas.**
>
> **A2:** We thank the reviewer for pointing this out. In Equation (1), the subscript $j$ denotes the index of a point within the superpoint, and the subscript $c$ corresponds to the element in the one-hot vector representing class $c$. We will supplement the main text with explicit definitions of these subscripts in the revised manuscript to ensure that the notation is clear and unambiguous.
>
> **Q1: Generality across different backbones.**
>
> **A3:** We appreciate the reviewer’s insightful question and suggestion. To further examine the generality of our method, we conducted additional experiments using several representative backbones, including MinkowskiUNet-14 [4], MinkowskiUNet-18 [4], MinkowskiUNet-34 [4], PointNet++ [5], and PTv2 [6]. These models span voxel-based, point-based, and hybrid architectures, allowing us to evaluate whether our framework can adapt to different feature learning paradigms.
>
> As shown in the table below, our method consistently improves both closed-set and open-set performance across all tested backbones. This demonstrates that the proposed approach is not tied to a specific architecture and can be applied broadly to different 3D segmentation models.
>
> |            | PN++ [5]            |              | MK-14 [4]            |              | MK-18 [4]           |              | MK-34 [4]           |              | PTv2 [6]            |              |
> |------------|------------------|--------------|------------------|--------------|------------------|--------------|------------------|--------------|------------------|--------------|
> |            | Source           | Ours         | Source           | Ours         | Source           | Ours         | Source           | Ours         | Source           | Ours         |
> | mIoU (%)   | 26.51            | **+6.80**        | 35.76            | **+7.00**        | 40.26            | **+4.83**        | 42.79            | **+4.80**        | 44.45            | **+4.27**        |
> | AUROC (%)  | 53.05            | **67.35**        | 64.90            | **75.03**        | 65.80            | **73.39**        | 63.16            | **72.15**        | 63.71            | **78.38**        |
> | FPR95 (%)  | 95.47            | **81.16**        | 85.30            | **76.85**        | 78.39            | **71.91**        | 87.70            | **79.66**        | 85.21            | **70.03**        |
>
>
> These results demonstrate the strong generality and compatibility of our method across diverse backbone architectures.
>
> > [4] Choy, Christopher, et al. "4d spatio-temporal convnets: Minkowski convolutional neural networks." CVPR 2019.
> >
> > [5] Qi, Charles, et al. "Pointnet++: Deep hierarchical feature learning on point sets in a metric space." NIPS 2017.
> >
> > [6] Wu, Xiaoyang, et al. "Point transformer v2: Grouped vector attention and partition-based pooling." NIPS 2022.

---

> ### Author Response · Authors · 2025-11-20
> **Responces to Reviewer sTXL [Part-2]**
>
> **Q2: mIoU improvement in Table 4.**
>
> **A4:** We sincerely appreciate your question. The modest mIoU improvement from the SR (Superpoint Representation) and SIDP (Superpoint ID Prototypes) modules stems from their design goal: enhancing unknown-class recognition. As such, their impact is most evident in open-set metrics like AUROC and FPR95, rather than closed-set segmentation accuracy. A more detailed ablation study in Table 8 shows the individual contributions of each module to closed-set mIoU. While the combined modules yield a clear improvement in closed-set performance, this gain is less apparent in Table 4, which primarily highlights open-set behavior.
>
> **Q3: Consideration of noisy samples when constructing prototypes.**
>
> **A5:** We sincerely appreciate your comments. Our method incorporates several mechanisms to address noisy samples during prototype construction. First, the Superpoint Representation (SR) module serves as an initial filtering stage. By aggregating geometrically coherent points into superpoints, SR reduces point-level noise and operates on more reliable regional representations rather than individual noisy points. Second, during prototype updates, we apply an EMA-based smoothing mechanism, which gradually integrates new information while suppressing abrupt shifts caused by incorrect pseudo-labels and stabilizing prototypes evolution over time.
>
>
> **Q4: Meaning of “superpoint representation (SR)” in Table 4.**
>
> **A6:** We apologize for the confusion and appreciate your helpful feedback. In Table 4, the term “superpoint representation (SR)” specifically indicates that we use the superpoint-level purity and confidence scores within the GMM-based separation module to distinguish known and unknown samples. In this stage, superpoints are first aggregated, and their purity and entropy-derived confidence values serve as inputs to the GMM, enabling category separation without involving the subsequent prototype-based refinement. We will revise the abbreviation "SR" in the manuscript to "SC (Superpoint Confidence)" to enhance the clarity of expression.

---

### Official Review · Reviewer_owHE · 2025-11-01

**Soundness:** 3
**Presentation:** 3
**Contribution:** 3
**Rating:** 6
**Confidence:** 3

**Summary:**

This paper addresses the challenging problem of open-set test-time adaptation (OSTTA) in 3D point cloud semantic segmentation, where models must adapt to unseen target domains containing both known and unknown categories. The authors identify a key limitation in existing OSTTA methods—most were designed for 2D image data and perform poorly on 3D point clouds due to geometric complexity and severe imbalance between in-distribution (ID) and out-of-distribution (OOD) samples. To address this, the paper proposes GOOD (Geometry-guided Out-of-Distribution Modeling), a novel framework that leverages geometric priors to cluster point clouds into superpoints, thereby transforming point-level imbalance into a structured representation. GOOD introduces (1) a Superpoint Representation Branch (SRB) with a Gaussian Mixture Model and prototype learning to distinguish ID and OOD superpoints, and (2) a Temporal Pseudo-label Branch (TPB) that enforces temporal consistency for reliable pseudo-label generation. The method is evaluated on four benchmarks, showing certain improvements over existing TTA and OSTTA baselines.

**Strengths:**

The paper presents a clear and well-motivated method that effectively extends OSTTA to 3D point clouds, with experimental validation and meaningful performance improvements.

**Weaknesses:**

1. The paper assumes access to reliable ego poses for temporal alignment, which might not hold in real-world scenarios—this assumption and its limitations are not discussed.

2. Prototypes are updated via EMA using pseudo-labeled superpoints. If the initial pseudo-labels are noisy, prototypes may drift toward incorrect centroids and subsequently reinforce wrong ID/OOD assignments. The paper does not analyze prototype stability over time or propose safeguards (re-weighting, outlier removal, confidence gating).

3. TPB depends on aggregating multiple frames via ego-pose alignment for K-NN temporal neighborhoods. In practice, pose noise (e.g., localization drift) or missing pose data will break temporal correspondence. The paper does not quantify sensitivity to alignment error nor propose how the method behaves when only single-frame inputs are available.

4. Several design choices (EMA coefficient α, cosine similarity threshold τ, use of Dice loss for ID pseudo-label learning) are empirically reasonable but not theoretically motivated. This weakens understanding of when and why the method will succeed or fail.

5. The paper mentions partitioning the point cloud into sub-regions and applying RANSAC to each, but does not specify sub-region division criteria or RANSAC hyperparameters. Only the original RANSAC literature is cited, and no task-specific settings for 3D point clouds are provided. This ambiguity makes the step impossible to replicate, which undermines the method’s practicality.

6. Appendix A.3.5 details the performance of different superpoint confidence metrics via Table 11 and concludes that "Purity+Entropy balances closed-set and open-set performance", but the main text (Section 3.2.2) only defines the confidence formula "C_k^sup=C_k^ent∙C_k^pur" and does not cite the appendix’s comparative results to support why "product" is better than other combinations. This makes the main text’s formula choice lack empirical support, and readers need to flip to the appendix to understand the rationale, disrupting the logical flow of the argument.

7. Eq.3 uses "z_i^t" to represent superpoint embedding, but the preceding text clearly defines "z_k^t" as the mean embedding of superpoint k—this confusion between "i" (point-level) and "k" (superpoint-level) makes readers unsure whether similarity is calculated per point or per superpoint.

**Questions:**

1. The SRB uses a GMM on purity and entropy. Have you experimented with other methods to avoid the Gaussian assumption? If so, why was GMM chosen?

2. The paper notes that prototypes are updated via EMA using pseudo-labeled superpoints, but does not quantify how prototype drift (caused by noisy initial pseudo-labels) affects long-term ID/OOD discrimination performance. Have you conducted experiments to measure prototype stability over consecutive test frames? Additionally, do you have strategies to mitigate drift, such as re-weighting pseudo-labels by superpoint confidence or removing outlier pseudo-labels before prototype update?

3. The TPB relies on multi-frame alignment via ego poses to build temporal neighborhoods, but real-world pose noise may corrupt this alignment. Have you quantified GOOD’s sensitivity to pose error (e.g., testing with artificial pose noise added to frames)? Specifically, how does pose noise impact the accuracy of temporal pseudo-labels and final segmentation performance?

4. Fig. 3 does not seem to be cited or mentioned in the main text. Perhaps a brief explanation can be given for it.

---

> ### Author Response · Authors · 2025-11-20
> **Responces to Reviewer owHE [Part-1]**
>
> We sincerely thank reviewer owHE for your thoughtful feedback on our work. We hope to address the concerns that have been raised. Unless otherwise specified, all experiments are conducted on Synth4D to SemanticKITTI. We will update our manuscript in a few days.
>
> **W1: Assumption with reliable ego poses for temporal alignment.**
>
> **A1:** We sincerely appreciate your constructive feedback, and we apologize for not discussing this assumption clearly in the original submission. In our framework, the use of ego poses follows the standard practice established in prior works [1,2]. Since relative pose estimation is a well-established problem in robotics and autonomous driving, it is generally feasible to obtain sufficiently stable pose estimates to support temporal association across consecutive scans. To verify the robustness of our framework under weakened pose conditions, we further replace the provided ego poses with a fast and accurate registration-based estimation using GICP [3]. GICP performs dense geometric alignment and can provide reliable relative poses at low computational cost. We compare its performance with the original setting that uses supplied poses, and the results are reported in the table below. The difference is minimal across all metrics, indicating that our method remains stable even when ego poses are replaced by registration-based estimates.
>
> In addition, as mentioned in Q3, we also provide results obtained without using any temporal alignment. That setting relies solely on geometric information within each LiDAR frame for adaptation. The combined results show that our framework is not limited by perfect pose availability and exhibits strong resilience across different alignment conditions.
>
> To enhance clarity in the revised manuscript, we will add a dedicated subsection discussing the limitations of this assumption, including practical constraints on pose availability and implications for real-world deployment. The paper will be updated within a few days.
>
> |            | Source  Model | with perfect pose | with pose estimated by GICP  |
> |------------|---------|--------|-------|
> | mIoU (%)   | 40.26   | +4.83  | +4.80 |
> | AUROC (%)  | 65.80   | 73.39  | 74.29 |
> | FPR95 (%)  | 78.39   | 71.91  | 71.23 |
>
> > [1] Saltori, Cristiano, et al. "Gipso: Geometrically informed propagation for online adaptation in 3d lidar segmentation." ECCV, 2022.
> >
> > [2] Zou, Tianpei, et al. "Hgl: Hierarchical geometry learning for test-time adaptation in 3d point cloud segmentation." ECCV, 2024.
> >
> > [3] Segal, Aleksandr, et al. "Generalized-icp." Robotics: science and systems. 2009.

---

> ### Author Response · Authors · 2025-11-20
> **Responces to Reviewer owHE [Part-2]**
>
> **W2/Q2: Prototype stability and safeguards.**
>
> **A2:** We thank the reviewer for highlighting the importance of prototype stability during adaptation. Since prototypes are updated via EMA using pseudo-labeled superpoints, it is indeed necessary to examine whether early-stage noise may lead to accumulated drift. To address this concern, we conducted an extensive comparison of several safeguard mechanisms, including weighted prototypes (WP)[4], threshold-based prototypes (TP) [5], and multi-center prototypes (MP) [6]. The results are reported in the table below.
>
> |            | Original  | WP          | TP      | MP         |
> |------------|-----------|-------------|---------|------------|
> | mIoU (%)   | +4.83     | +4.81       | +4.57   | **+4.94**  |
> | AUROC (%)  | 73.39     | **74.01**   | 72.87   | 72.72      |
> | FPR95 (%)  | 71.91     | **71.38**   | 75.76   | 76.40      |
>
> The results show that (1) the weighted prototype strategy achieves performance comparable to our original approach; (2) the threshold-based strategy performs slightly worse, likely due to the strict filtering of valuable samples. (3) the multi-center strategy shows mixed behavior, improving certain metrics while hurting others. We believe that the limited benefit of TP and MP stems from the nature of our prototype construction. Since the prototype is built from aggregated superpoints, and a Gaussian Mixture Model (GMM) is applied to select only high-confidence superpoints, the influence of noisy samples is already mitigated at an early stage. Additional safeguard mechanisms introduce extra hyperparameters that require further tuning, but they do not provide clear gains under our setting. Therefore, we retain the original prototype formulation for both effectiveness and simplicity.
>
> To further examine temporal stability, we also report the differences in classification accuracy for known and unknown categories before and after prototype updates, aggregated every 500 training steps. The values are: [1.61%, 1.85%, 2.57%, 2.37%, 3.83%, 1.41%, 2.52%, 4.86%]. These results show an overall upward trend, indicating that as training progresses, prototype updates tend to stabilize overall and reduce category assignment errors rather than reinforcing early-stage noise. These experiments support our conclusion that the baseline prototype mechanism is already stable and sufficiently robust.
>
> > [4] Manna, Siladittya, et al. "Correlation Weighted Prototype-Based Self-supervised One-Shot Segmentation of Medical Images." ICPR, 2024.
> >
> > [5] Su, Binyi, et al. "Toward generalized few-shot open-set object detection." TIP, 2024.
> >
> > [6] Qu, Sanqing, et al. "Bmd: A general class-balanced multicentric dynamic prototype strategy for source-free domain adaptation." ECCV, 2022.

---

> ### Author Response · Authors · 2025-11-20
> **Responces to Reviewer owHE [Part-3]**
>
> **W3/Q3: Sensitivity to alignment error and single-frame availability.**
>
> **A3:** We appreciate the reviewer’s insightful question regarding the reliance of the Temporal Pseudo-label Branch (TPB) on ego-pose alignment and the potential effects of pose noise or missing pose information.
>
> To quantify sensitivity, we conducted a detailed pose-noise analysis by injecting controlled translation and rotation perturbations into the ego poses. As shown in the first table below, translation noise is denoted as $T_{err}$, where the noise is uniformly sampled within $[−X, X]$ meters, and rotation noise is denoted as $R_{err}$, sampled within $[−Y, Y]$ degrees.
>
> |            | Source Model  | with perfect pose | $T_{0.5}$-$R_{0}$ | $T_{1.0}$-$R_{0}$ | $T_{2.0}$-$R_0$ | $T_{0.0}$-$R_{5}$ | $T_{0.0}$-$R_{10}$ | $T_{0.0}$-$R_{20}$ | $T_{0.5}$-$R_{5}$ | $T_{1.0}$-$R_{10}$ | $T_{2.0}$-$R_{20}$ |
> |------------|---------|--------|---------|---------|---------|---------|----------|----------|---------|----------|----------|
> | mIoU (%)   | 40.26   | **+4.83**  | +4.54   | +4.29   | +2.86   | +1.75   | +0.91    | +0.51    | +0.98   | +0.55    | -0.08    |
> | AUROC (%)  | 65.80   | **73.39**  | 73.28   | 73.94   | 74.53   | 74.43   | 72.27    | 72.76    | 73.31   | 72.41    | 70.35    |
> | FPR95 (%)  | 78.39   | **71.91**  | 73.55   | 74.92   | 72.39   | 68.95   | 77.52    | 74.21    | 75.11   | 78.05    | 77.84    |
>
> From these results, we observe that rotation noise has a more substantial impact on performance than translation noise. When both translation and rotation errors become large (for example, 2 meters and 20 degrees), the temporal alignment deteriorates to the point where the adaptation effect is almost entirely lost, which aligns with expectations for correspondence-driven temporal regularization.
>
> To account for practical scenarios where registration algorithms or accurate poses are unavailable, we also evaluate a single-frame variant of our method. In this setting, we remove the temporal consistency component and perform only intra-frame superpoint aggregation and pseudo-labeling. The second table shows that, although performance declines relative to the full GOOD pipeline, the closed-set mIoU of GOOD-single still exceeds that of GIPSO, and the open-set metrics (AUROC and FPR95) also remain competitive or even improved. This demonstrates that our framework maintains meaningful adaptation capability even without temporal aggregation.
>
> |            | Source Model  | GOOD       | GOOD-single | HGL       | GIPSO     |
> |------------|---------|------------|-------------|-----------|-----------|
> | mIoU (%)   | 40.26   | **+4.83**      | +2.37       | +2.90     | +2.08     |
> | AUROC (%)  | 65.80   | **73.39**      | 73.23       | 64.40     | 64.47     |
> | FPR95 (%)  | 78.39   | **71.91**      | 75.04       | 79.82     | 80.40     |
>
> Overall, these analyses clarify the behavior of our method across different alignment conditions and show that GOOD is not critically dependent on perfect ego-poses. When alignment is accurate, temporal aggregation provides gains, while in the absence of reliable poses, the single-frame variant remains a feasible and robust alternative.

---

> ### Author Response · Authors · 2025-11-20
> **Responces to Reviewer owHE [Part-4]**
>
> **W4: Rationale behind several design choices.**
>
> **A4:** We thank the reviewer for pointing out the need to clarify the rationale behind several design components. While our choices are primarily guided by empirical behavior, we conducted extensive analyses to better understand their effects and ensure stable performance.
>
> First, the hyperparameter analyses for the EMA coefficient $\alpha$ and the cosine similarity threshold $\tau$ are already provided in Supplementary Table 9 and Figure 5(d) of the main text, respectively. These experiments show that the method is not overly sensitive to the selection of $\alpha$ or $\tau$ within a reasonable range, which supports the robustness of these design choices even in the absence of a strong theoretical prior.
>
> Second, we performed additional experiments comparing multiple loss functions for ID pseudo-label learning, including SCE loss, CE loss, Dice loss, and Soft Dice loss. The results are summarized in the table below. Soft Dice loss introduces a severe degradation in open-set detection capability, as seen from its significantly lower AUROC and higher FPR95. Conversely, SCE loss and CE loss produce strong open-set results but insufficient closed-set segmentation performance, indicating that these losses overemphasize separability at the cost of reconstruction quality.
>
> Dice loss provides the best balance across both closed-set and open-set tasks. Dice loss is known to be particularly effective for small objects and boundary regions, and its inherent normalization helps mitigate class imbalance, which is especially relevant for superpoint aggregation and prototype construction. These properties explain why Dice loss supports both stable pseudo-label refinement and consistent adaptation across domains.
>
> |            | Source  | Dice Loss (GOOD) | Soft Dice Loss | SCE Loss | CE Loss |
> |------------|---------|-----------|----------------|----------|---------|
> | mIoU (%)   | 40.26   | **+4.83**     | +4.81          | +3.33    | +3.53   |
> | AUROC (%)  | 65.80   | 73.39     | 68.62          | **76.70**    | 75.70   |
> | FPR95 (%)  | 78.39   | 71.91     | 84.87          | **63.26**    | 64.07   |
>
> Taken together, these analyses demonstrate that although the design choices are empirically driven, they are grounded in well-understood properties of temporal smoothing, geometric similarity, and segmentation loss behavior. We will include additional explanations in the revised manuscript to improve clarity regarding when and why these components contribute to stable adaptation.
>
> **W5: Replicability of point cloud partitioning and RANSAC settings.**
>
> **A5:** We thank the reviewer for pointing out the ambiguity regarding the ground plane extraction procedure. Our intention is to make this preprocessing step fully replicable, and we will clarify the complete configuration in the revised manuscript. For all datasets, we divide each LiDAR scan into 10×10 sub-regions. This partitioning is used to reduce the effect of uneven ground surfaces and to ensure that RANSAC is applied on more locally consistent regions, which increases the accuracy of plane estimation.
> To avoid fitting non-ground planes, we define candidate ground points as those with height values below $-1m$ relative to the ego-vehicle coordinate system. This simple threshold is effective across datasets due to the similar height of vehicle-mounted LiDAR sensors. For the RANSAC procedure, we set the distance threshold to 0.1, the minimum number of points required for model fitting (n) to 3, and the number of iterations to 1000. We will include a detailed description of the above criteria and parameters in the revised manuscript.
> **In addition, all code will be released after finalization of the paper, ensuring that the entire preprocessing pipeline can be reproduced precisely.**
>
> **W6: Missing empirical justification for confidence formulation in main text.**
>
> **A6:** Thank you for your careful reading and helpful suggestion. We agree that the current main text lacks an explicit reference to the comparative results presented in Appendix A.3.5, which may hinder the clarity and completeness of the argument. In the revised manuscript, we will explicitly cite Table 11 from the appendix in Section 3.2.2 and state that multiple confidence combinations. As shown in that table, the product-based metric ("Purity × Entropy") consistently offers the best trade-off between closed-set mIoU and open-set AUROC/FPR95, justifying its use as the default setting in our framework.
>
> **W7: Formaulation typo.**
>
> **A7:**  Thank you for pointing this out. We acknowledge the notation inconsistency, and we will correct the typo $z_i^t$ to the correct form $z_k^t$ in Equation (3) to maintain notational coherence across the description of superpoint-wise operations.

---

> ### Author Response · Authors · 2025-11-20
> **Responces to Reviewer owHE [Part-5]**
>
> **Q1: Exploration beyond GMM strategy.**
>
> **A8:** Thank you for this question. We investigate alternative strategies that do not rely on the Gaussian assumption. In particular, we evaluate a threshold-based strategy that directly separates known and unknown categories using manually selected thresholds on purity and entropy. The table below includes two threshold variants: threhold-open, which favors open-set performance, and thresholding-closed, which favors closed-set performance.
>
> Although these methods achieve reasonable results, they require additional hyperparameter tuning to balance open-set and closed-set behavior. More importantly, the optimal thresholds vary substantially across datasets, making it difficult to maintain stable performance without dataset-specific adjustment. In contrast, the GMM-based method adapts automatically to the score distribution of each domain and does not require manual threshold selection. For this reason, we retain the GMM-based division as our default choice.
>
> |            | Source  | GOOD w/ GMM         | GOOD w/ threhold-open   |  GOOD w/ threhold-closed  |
> |------------|---------|-------------|-------|-------|
> | mIoU (%)   | 40.26   | +4.83       | +4.43  | **+4.90**  |
> | AUROC (%)  | 65.80   | 73.39       | **74.46** | 73.77 |
> | FPR95 (%)  | 78.39   | 71.91       | **70.55** | 73.01 |
>
> **Q2: Prototype stability.**
>
> **A9:** Please refer to **A2**.
>
> **Q3: Robustness of TPB branch against pose noise.**
>
> **A10:** Please refer to **A3**.
>
> **Q4: Figure 3 lacks explanation.**
>
> **A11:** Thank you for pointing this out. We will make the correction in the revised manuscript.

---

### Comment · Area_Chair_cJm8 · 2025-11-27
**Reminder: Engage in Discussions and Finalize Your Rating**

Dear Reviewers,

Thank you for your valuable reviews. With the Reviewer-Author Discussions deadline approaching, please take a moment to read the authors’ rebuttal and the other reviewers’ feedback, and participate in the discussions and respond to the authors. Finally, be sure to complete the “Final Justification” text box and update your “Rating” as needed. Your contribution is greatly appreciated. I will flag irresponsible (final) reviews and/or any reviewers not participating in discussions.

Reviewers are expected to stay engaged in discussions, initiate them, respond to authors’ rebuttal, ask questions, and listen to answers to help clarify remaining issues.

It is not OK to stay quiet.

It is not OK to leave discussions till the last moment.

If authors have resolved your (rebuttal) questions, do tell them so.

If authors have not resolved your (rebuttal) questions, do tell them so too.

Thanks,

AC

---

### Author Response · Authors · 2025-11-27

We sincerely thank the Area Chair cJm8 and all reviewers for their time, constructive comments, and valuable suggestions. We have carefully addressed all raised issues in our detailed responses and have revised the manuscript accordingly. A highlighted version of the updated paper has been provided to clearly indicate the changes. We hope that these clarifications and improvements adequately resolve the concerns and help convey the contributions of our work more clearly.

---

### Author Response · Authors · 2025-12-01
**Summary comments to Area chair**

We sincerely thank the Area Chair and reviewers for their time and constructive feedback, which helped clarify the motivation, contributions, and empirical findings of our work. Below is a concise summary of our contributions, positive feedback, and responses to key concerns.

**(1) Our contributions:**

We systematically study open-set test-time adaptation for 3D point cloud semantic segmentation (OSTTA-3DSeg), with main contributions:
*  To the best of our knowledge, we are the first to formulate the open-set test-time adaptation problem for OSTTA-3DSeg.
* We propose Geometry-guided Out-of-Distribution Modeling (GOOD), which shifts the identification from point-level to superpoint-level, leading to more robust and coherent segmentation results.
* GOOD can integrate with existing TTA-3Dseg methods, significantly improving open-set performance.


**(2) Summary of Positive Feedback from Reviewers:**

* Clear motivation and sound design: SRB and TPB effectively address the ID–OOD imbalance (owHE, XJDX).
* Innovative approach: Superpoint clustering and temporal pseudo-labeling provide a practical way to distinguish ID and OOD points (owHE, sTXL).
* Strong empirical validation: Experiments and visualizations demonstrate consistent performance gains (owHE, sTXL, XJDX).

**(3) Summary of the reviewers' key suggestions and concerns**

* Ego-Pose Assumption and Robustness Analysis (owHE)

Our framework assumes access to ego poses for temporal alignment. Reviewer owHE suggested analyzing robustness under imperfect poses. We evaluated GOOD using registration-based pose estimates (GICP) and controlled translation/rotation noise. Results show temporal aggregation remains effective, with only significant rotation + translation perturbations degrading performance (e.g., 2 m and 20°).

|            | Source  Model | with perfect pose | with pose estimated by GICP  |
|------------|---------|--------|-------|
| mIoU (%)   | 40.26   | +4.83  | +4.80 |
| AUROC (%)  | 65.80   | 73.39  | 74.29 |
| FPR95 (%)  | 78.39   | 71.91  | 71.23 |

Even without temporal cues, GOOD-single surpasses baselines, demonstrating robustness.

|            | Source Model  | GOOD       | GOOD-single | HGL       | GIPSO     |
|------------|---------|------------|-------------|-----------|-----------|
| mIoU (%)   | 40.26   | **+4.83**      | +2.37       | +2.90     | +2.08     |
| AUROC (%)  | 65.80   | **73.39**      | 73.23       | 64.40     | 64.47     |
| FPR95 (%)  | 78.39   | **71.91**      | 75.04       | 79.82     | 80.40     |


* Generality across different backbones (sTXL)

Reviewer sTXL recommended testing different backbones. We tested MinkowskiUNet-14/18/34, PointNet++, and PTv2, covering voxel-, point-, and hybrid-based architectures. GOOD consistently improves closed- and open-set performance, confirming strong generality.

|            | PN++            |              | MK-14            |              | MK-18          |              | MK-34           |              | PTv2            |              |
|------------|------------------|--------------|------------------|--------------|------------------|--------------|------------------|--------------|------------------|--------------|
|            | Source           | Ours         | Source           | Ours         | Source           | Ours         | Source           | Ours         | Source           | Ours         |
| mIoU (%)   | 26.51            | **+6.80**        | 35.76            | **+7.00**        | 40.26            | **+4.83**        | 42.79            | **+4.80**        | 44.45            | **+4.27**        |
| AUROC (%)  | 53.05            | **67.35**        | 64.90            | **75.03**        | 65.80            | **73.39**        | 63.16            | **72.15**        | 63.71            | **78.38**        |
| FPR95 (%)  | 95.47            | **81.16**        | 85.30            | **76.85**        | 78.39            | **71.91**        | 87.70            | **79.66**        | 85.21            | **70.03**        |


* Comparison with open-vocabulary 3D segmentation methods (XJDX)

Following reviewer XJDX’s suggestion, we evaluated OpenScene, a representative open-vocabulary 3D segmentation model, in our OSTTA setting (nuScenes → SemanticKITTI). The overall mIoU dropped to 8%, with only the “road” category achieving 25%, while all other categories remained uniformly low. This limitation stems from: (1) modality gap between 2D vision–text alignment and 3D-only data; (2) single-dataset training/evaluation limiting cross-domain generalization; (3) suboptimal performance on nuScenes compared to fully supervised baselines.; and (4) heavy dependence on text-embedding fine-tuning (42% mIoU with tuning vs. 30% without tuning).

We also examined 3DOSSS as an alternative open-set/open-vocabulary method (see Supplementary Material A.3.6). Importantly, GOOD is complementary and can enhance cross-domain generalization of open-vocabulary methods by providing adaptation at test time.

---

### Meta-Review · Area_Chair_p9D3 · 2026-01-04

**Summary:**

Three reviewers assessed this submission, all initially recommending borderline acceptance. The primary concerns focused on the robustness of the method in real-world scenarios, specifically its reliance on accurate ego-poses for temporal alignment and the stability of prototype updates under noisy pseudo-labels. Reviewers also requested more comprehensive computational metrics (GFLOPs), broader backbone generalization experiments, and a clearer distinction between this test-time adaptation approach and existing open-vocabulary segmentation models like OpenScene. The authors provided a compelling rebuttal that included sensitivity analyses for pose errors, additional efficiency benchmarks, and a clear articulation of how their method complements open-vocabulary paradigms. Given the thorough resolution of technical and conceptual issues, the AC recommends acceptance.

**Reviewer Concerns:**

The significant technical concerns regarding the method's dependency on reliable ego-poses and temporal alignment, raised extensively by Reviewer owHE, were fully addressed through new experiments quantifying sensitivity to pose noise. Similarly, the shared apprehension from Reviewers owHE and sTXL regarding the risk of prototype drift due to noisy initial pseudo-labels was resolved by the authors' stability analysis and clarification of safeguards. On the experimental side, Reviewer sTXL’s requests for GFLOPs and tests with different backbones were met with supplementary data proving the method's efficiency and generality. Finally, Reviewer XJDX’s inquiry into the positioning of the work against open-vocabulary segmentation models was effectively answered; the authors clarified that their proposal serves as a complementary adaptation module that enhances the cross-domain generalization of architectures like OpenScene, rather than competing with them.

**Reviewer Scores:**

Reviewer owHE is expected to maintain his/her positive score (6) or increase it to an 8. The authors provided a rigorous response to the reviewer's specific concerns about pose sensitivity and theoretical motivations, offering the quantification and safeguards necessary to substantiate the method's robustness. Similarly, Reviewer sTXL is likely to maintain his/her score or raise it, given that the missing computational metrics and backbone experiments were provided in full. Reviewer XJDX is also expected to maintain or increase their score to an 8, as the conceptual ambiguity regarding the method's relationship to open-vocabulary segmentation was resolved.

---

### Decision · Program_Chairs · 2026-01-26

Accept (Poster)